# Chemically induced mutations in a MutaMouse reporter gene inform mechanisms underlying human cancer mutational signatures

Marc A. Beal[1,3,4], Matthew J. Meier[1,4], Danielle P. LeBlanc[1], Clotilde Maurice [1,3], Jason M. O'Brien[2], Carole L. Yauk [1] & Francesco Marchetti [1✉]

Transgenic rodent (TGR) models use bacterial reporter genes to quantify in vivo mutagenesis. Pairing TGR assays with next-generation sequencing (NGS) enables comprehensive mutation pattern analysis to inform mutational mechanisms. We used this approach to identify 2751 independent *lacZ* mutations in the bone marrow of MutaMouse animals exposed to four chemical mutagens: benzo[a]pyrene, *N*-ethyl-*N*-nitrosourea, procarbazine, and triethylenemelamine. We also collected published data for 706 *lacZ* mutations from eight additional environmental mutagens. We report that *lacZ* gene sequencing generates chemical-specific mutation signatures observed in human cancers with established environmental causes. For example, the mutation signature of benzo[a]pyrene, a carcinogen present in tobacco smoke, matched the signature associated with tobacco-induced lung cancers. Our results suggest that the analysis of chemically induced mutations in the *lacZ* gene shortly after exposure provides an effective approach to characterize human-relevant mechanisms of carcinogenesis and propose novel environmental causes of mutation signatures observed in human cancers.

[1] Environmental Health Science and Research Bureau, Healthy Environments and Consumer Safety Branch, Health Canada, Ottawa, Ontario K1A 0K9, Canada. [2] National Wildlife Research Centre, Environment and Climate Change Canada, Ottawa, ON K1A 0H3, Canada. [3] Present address: Existing Substances Risk Assessment Bureau, Health Canada, Ottawa, ON, Canada. [4] These authors contributed equally: Marc A. Beal, Matthew J. Meier. ✉email: francesco.marchetti@canada.ca

Transgenic rodent (TGR) mutation reporter models have enabled unprecedented insights into spontaneous and chemically induced mutagenesis[1]. Studies of over 200 chemicals, including more than 90 carcinogens, have demonstrated that TGR models offer high sensitivity and specificity for identifying mutagenic carcinogens[1,2]. One of the most commonly used TGR models is the MutaMouse whose genome was recently sequenced[3]. The MutaMouse harbors ~29 copies of the bacterial *lacZ* transgene on each copy of chromosome 3[4]. This is a neutral, transcriptionally-inert reporter gene carried on a shuttle vector that can be recovered from any cell type and transfected into a bacterial host to detect somatic or germline mutations that occurred in vivo[5,6]. A major advantage of TGR models is the possibility to sequence mutants in order to characterize mutation patterns. This information is necessary to understand mutational mechanisms associated with mutagen exposure and response in different tissues, life stages, genetic backgrounds or other contexts. Advances in next-generation sequencing (NGS) technologies have enabled rapid and accurate characterization of TGR mutants[7,8], and integrated TGR–NGS approaches have been used to sequence thousands of mutations[8,9] at a fraction of the cost of whole-genome sequencing. Thus, TGR–NGS approaches currently provide a unique methodology for simultaneously assessing the magnitude of the mutagenic response and characterizing mutations to inform underlying mechanisms.

Somatic mutation analysis by NGS has greatly advanced our understanding of the mutational processes operating in human cancers. Algorithms have been developed to mine the extensive database of single nucleotide variations (SNVs) in cancer genomes to identify mutational signatures contributing to individual cancers[10–12]. These signatures represent a computationally derived prediction of the relative frequencies of mutation types induced by processes that contribute to all observed mutations within The Cancer Genome Atlas datasets (TCGA; https://www.cancer.gov/about-nci/organization/ccg/research/structural-genomics/tcga). As opposed to standard mutation characterization that simply describes the frequency of individual nucleotide changes, mutational signatures incorporate flanking nucleotide context. Originally, 30 mutational signatures from 40 different cancer types were identified and reported in the Catalogue of Somatic Mutations in Cancer (COSMIC) database[13,14]. This database was recently expanded to include 71 cancer types and 77 signatures, including 49 single base substitution (SBS) signatures, 11 doublet base substitution signatures, and 17 small insertion and deletion (indel) signatures[15]. Each SBS signature encompasses 96 possible mutation types (i.e., 6 possible base-pair alterations × 4 different 5' bases × 4 different 3' bases). Many of these signatures have been attributed to endogenous processes, but chemical mutagens also play a major contributing role in certain signatures[16]. For example, SBS 4 signature is observed in lung cancer and has been attributed to tobacco smoke[16,17]. This signature has been recapitulated by exposing murine embryo fibroblasts to benzo[a]pyrene (BaP)[18,19], a major mutagenic component of tobacco smoke. However, several of the mutational signatures currently have no known endogenous or exogenous causative agents[17]; thus, identification of exogenous environmental exposures that contribute to these mutational signatures may aid in elucidating carcinogenic mechanisms.

The pattern of mutations observed in a fully developed cancer is a composite of the signature of the molecular initiating events in the early stages of tumor formation and signatures arising as a result of genomic instability in the evolving tumor[20]. For example, a tumor that originates in the lung of a smoker will have a mutational fingerprint that is caused primarily by DNA damage induced by the many mutagenic compounds found in tobacco smoke[21]. In addition, the person's age at the time of tumor formation will also determine the contribution of "clock-like" signatures, caused by lifetime DNA replication, to the fingerprint of the tumor[22]. There is now compelling evidence that analysis of the pattern of mutations in a cancer can provide clues to past environmental exposures that contributed to the development of the cancer[23,24]. Implicit in this is that the exposure signature should be present in the normal tissue before the carcinogenic process becomes apparent. Indeed, previous studies have demonstrated that mutational signatures observed in aflatoxin-induced cancers are observed in normal tissues long before tumor formation[25,26]. Recent work in vivo[27] and in vitro[28] has shown that chemical-specific signatures detected shortly after exposure in non-tumor target tissues match signatures seen in human cancers. Thus, characterization of short-term mutational signatures in non-tumor tissues is a valuable approach to elucidate human-relevant mechanisms of carcinogenesis.

In this study, we used TGR–NGS to characterize mutations induced by four established mutagens to determine if these mutation profiles inform carcinogenic mechanisms within COSMIC signatures. For this purpose, we chose four chemicals with varying mutagenic potencies, mode of action, and carcinogenic classification (as determined by the International Agency for Research on Cancer): one known class 1 carcinogen, BaP; two probable class 2 carcinogens including *N*-ethyl-*N*-nitrosourea (ENU) and procarbazine (PRC); and one class 3 chemical with inadequate information to be classified, triethylenemelamine (TEM). MutaMouse males were exposed by gavage to the chemicals or solvent for 28 days and DNA was collected from bone marrow for analysis. Bone marrow was chosen as the tissue to study because it is one of the most commonly used tissues for mutagenicity assessment for regulatory purposes. To further compare *lacZ* mutation patterns and COSMIC signatures, published Sanger sequencing data from 17 studies involving eight mutagens were also examined (Supplementary Table 1). These studies include data from mice exposed to electromagnetic radiation[29–33], alkylating agents and adduct-forming agents[34–40], and a nitrogenous base analog[41]. Data from control animals in these studies and others[42–46] were also included to generate a background mutation signature. Using *lacZ*-derived mutation data, we validated COSMIC signatures with proposed aetiologies through the identification of the expected signatures in the relevant exposure groups. We argue that analysis of COSMIC signatures observed in exposed animals can be used to generate or test hypotheses of mutagenic mechanisms associated with human mutational signatures of unknown etiology.

## Results

**Experimental approach and mutant frequencies**. We used mutation patterns generated in-house for four chemicals (BaP, ENU, PRC, and TEM) and vehicle-matched controls, and published data from eight agents, including BaP and ENU (Supplementary Table 1) and their matched controls, to query the COSMIC database and elucidate the role of environmental mutagens in cancer development. The overall experimental design is summarized in Fig. 1.

Mutation patterns were generated from plaques collected during experiments aimed at evaluating the induction of mutations in the bone marrow of MutaMouse males exposed to either BaP, ENU, PRC, or TEM using the *lacZ* assay[5]. Mutant frequencies were previously reported for BaP[8], PRC[47], and TEM[48]. All of the exposures caused increases in mutant frequencies relative to vehicle-matched controls (Supplementary Table 2), and the results were highly significant ($P < 0.0001$) for BaP (122.9-fold), PRC (9.7-fold), and ENU (7.2-fold). TEM exposure also increased mutant frequency relative to controls

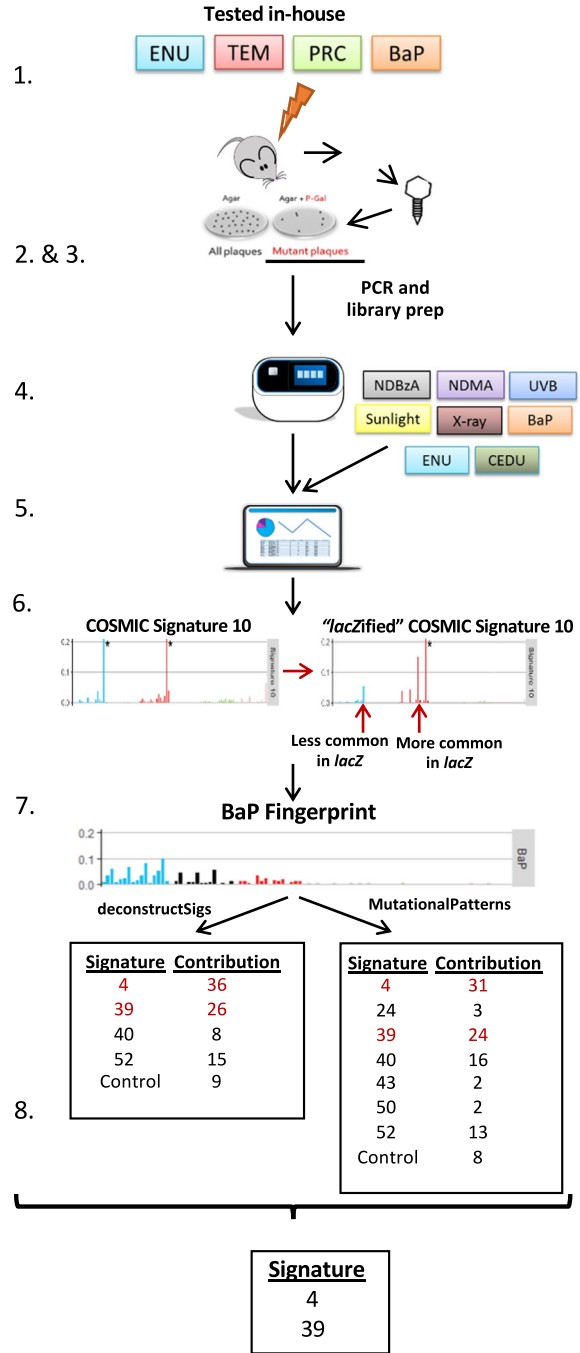

**Tested in-house**

1.

2. & 3.

PCR and
library prep

4.

5.

6.

COSMIC Signature 10     "lacZified" COSMIC Signature 10

Less common          More common
in *lacZ*              in *lacZ*

7.

**BaP Fingerprint**

deconstructSigs                    MutationalPatterns

| Signature | Contribution |
|-----------|--------------|
| 4 | 36 |
| 39 | 26 |
| 40 | 8 |
| 52 | 15 |
| Control | 9 |

| Signature | Contribution |
|-----------|--------------|
| 4 | 31 |
| 24 | 3 |
| 39 | 24 |
| 40 | 16 |
| 43 | 2 |
| 50 | 2 |
| 52 | 13 |
| Control | 8 |

8.

| Signature |
|-----------|
| 4 |
| 39 |

**COSMIC signatures that best describe the BaP mutational fingerprint**

**Fig. 1 Experimental design.** The experimental workflow included: animal exposure and determination of mutant frequencies (steps 1–2); sequencing of collected plaques and collection of published *lacZ* sequenced data (steps 3–4); generation of mutation profiles (steps 5–6); and query of the COSMIC database to identify mutational signatures that contributed to the mutation profile of tested agents (steps 7–8). The steps are detailed here and numbered as in the figure: (1) Four chemicals were tested in-house against solvent controls using the TGR in vivo mutagenicity assay. (2) Mutant plaques from controls and chemical-exposed mice were collected and pooled per individual. (3) Mutant plaques were PCR amplified as two technical replicates, library prepped and sequenced on the Ion Proton Platform. SNVs were called and corrected for clonal expansion. (4) Published Sanger sequencing data were compiled for eight additional mutagens, plus controls, tested using the *lacZ* plasmid or MutaMouse mice. (5) All sequencing data (Sanger and Ion Proton) were imported into the R console and trinucleotide mutation context were obtained using the "mutationContext" function. (6) To compare human COSMIC signatures and *lacZ* mutation data, the COSMIC signatures were normalized to *lacZ* trinucleotide frequencies and each of the 96 trinucleotide substitutions were represented as relative frequency. (7) The "deconstructSigs" and "MutationalPatterns" packages were used in parallel to identify COSMIC signatures that best describe the mutational fingerprint of mutagen exposure. (8) High confidence signatures were selected as those that: (i) were detected by both "deconstructSigs" and "MutationalPatterns"; (ii) contributed at least 20%; (iii) had a cosine similarity of 0.5 or higher with the mutational fingerprint.

the two sequencing approaches were consistent for each of the three groups. Thus, within each group, the two sets of mutations were combined. Overall, there were 1046, 2914, 129, 902, and 428 mutants sequenced in the Controls, BaP, PRC, ENU, and TEM groups, respectively.

In the *lacZ* gene, there are 3096 positions × 3 possible substitutions at each position for a total of 9288 possible unique SNV events; however, not all of these can be detected using a functional assay, since many result in silent mutations. Sequencing mutants from the different groups identified 891 unique SNVs, 338 of which overlapped between two or more groups (Supplementary Fig. 1). Specific to each group, there were 55, 377, 14, 85, and 22 unique SNVs for Controls, BaP, PRC, ENU, and TEM, respectively (Supplementary Table 3). The mutations detected in this study are limited almost exclusively to point mutations and small indels (1–21 bp), as large deletions are infrequently recovered during packaging of the DNA for the *lacZ* assay[8].

The mutation patterns of the four chemicals were significantly different from the control mutation pattern (Fig. 2; $P \le 0.0008$; Supplementary Data 1). The COSMIC convention is to represent mutations based on pyrimidine changes; thus, we present our mutation patterns using the same convention. The main spontaneous mutation is represented by C>T transitions, which are thought to arise through spontaneous mechanisms such as deamination of methylated cytosines[49]. Although there may be proportional declines in specific mutations relative to controls (Fig. 2), all of the chemicals tested in this study, with the exception of TEM, increased the mutation frequency of substitutions (e.g., C>T; Supplementary Fig. 2).

The mutation patterns of BaP and ENU are consistent with previous observations. BaP exposure caused cytosine transversions and indels (Fig. 2), mainly C>A SNVs, consistent with the formation of bulky DNA adducts mostly at the N2 of guanine[8]. ENU induced T>A mutations consistent with alkylation of thymine, specifically O2- and O4-ethyl thymine[50,51]. We found that PRC induced T>A mutations and, to a lesser extent T>C

(1.6-fold; $P = 0.048$), but it was less potent than the other agents. The potency ranking of exposures (BaP > PRC/ENU > TEM) was consistent with expectations.

**Mutation characterization and pattern analyses.** NGS of 5419 mutant plaques from bone marrow DNA enabled the characterization of 2751 independent mutations that were distributed as follows: 512, 1547, 120, 419, and 153 for controls, BaP, PRC, ENU, and TEM, respectively (Supplementary Table 3). Sequenced plaques from BaP, ENU, and controls were generated by both NGS and Sanger sequencing. Specifically, there were 60, 207, and 508 independent mutations identified by Sanger for BaP, ENU, and controls, respectively. The mutation patterns generated by

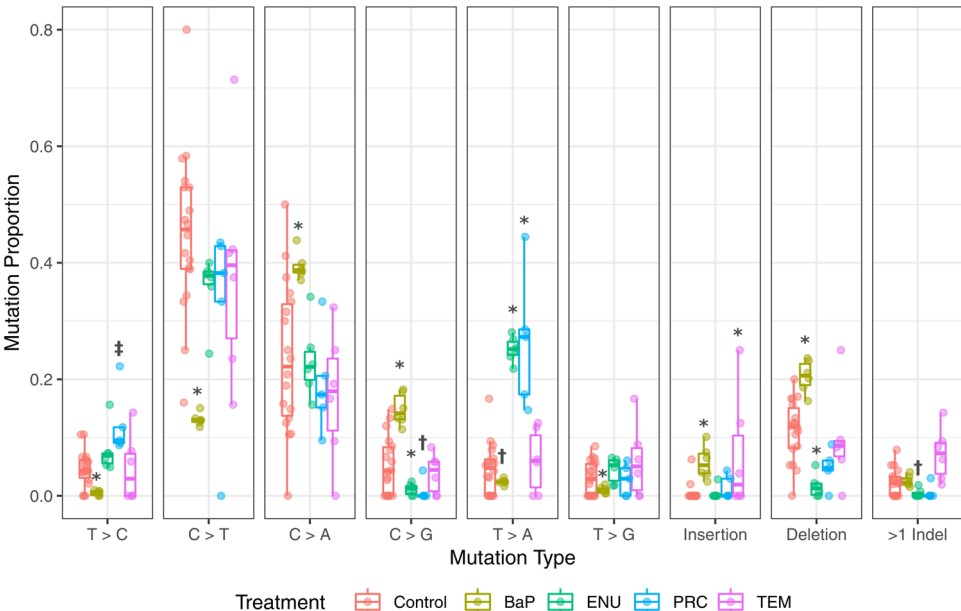

**Fig. 2 Spontaneous and chemical-induced mutation proportions in bone marrow as characterized by NGS.** BaP, shown in yellow, has significantly higher proportions of C>A, C>G, insertions, and deletions compared to control (red). In contrast, there is a lower proportion of T>C, C>T, T>A, and T>G mutations than control. ENU, shown in green, has a higher proportion of T>A mutations, while C>T, C>G, and deletions are lower. PRC, shown in blue, has a higher proportion of T>A compared to control, and a marginally significant increase in T>C mutations compared to control ($P = 0.055$). The mutation pattern for TEM, shown in purple, is most similar to that of the control, with the exception of a significant increase in the proportion of insertions. ‡$P < 0.1$, †$P < 0.05$, *$P < 0.0001$. The number of animals for controls, BaP, ENU, PRC, and TEM were, 18, 6, 6, 5, and 6, respectively.

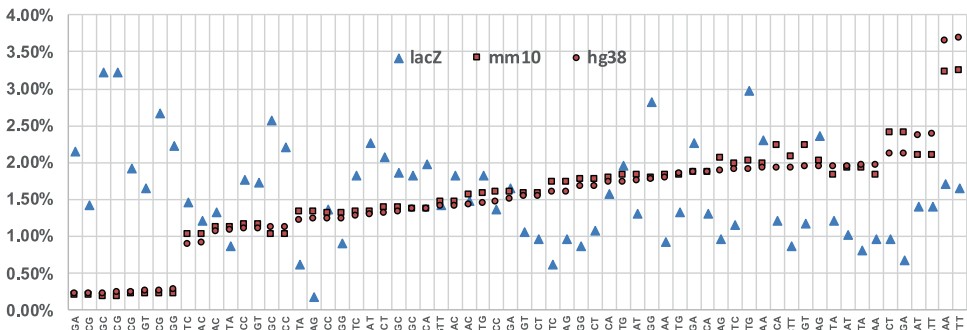

**Fig. 3 Trinucleotide context differences between the *lacZ* transgene, mouse genome, and human genome.** Comparison of the frequencies of the 64 possible trinucleotides among the *lacZ* transgene (lacZ), mouse genome (mm10), and human genome (hg38) show that mouse and human genome frequencies are comparable with each other, while *lacZ* is more variable and biased towards some GC rich trinucleoties.

mutations, which is consistent with the pattern of mutations that was observed in an endogenous gene[52]. The mutation pattern of TEM was significantly different from controls, but there were no significant changes in specific SNV types. Instead, the effect is mainly driven by the higher proportion of TEM-induced single nucleotide insertions compared to control animals. TEM also induced the highest proportion of >1 bp indels among all chemicals tested (Fig. 2).

**Identification of COSMIC signatures using *lacZ* mutations.** We explored the use of the *lacZ* sequence to obtain mutational signatures associated with human cancers. Although the COSMIC database (version 3) also includes doublet base substitution and indel signatures, we focused on SBS signatures because the *lacZ* assay detects almost exclusively these types of events. Because the COSMIC database is based on a much larger dataset of mutations than the available *lacZ* mutations, we first divided each trinucleotide frequency in the *lacZ* transgene (Fig. 3; Supplementary

Data 2) by the respective human genome frequencies (hg38) to create a *lacZ*-normalized set of the 49 COSMIC SBS signatures (Supplementary Fig. 3 and Supplementary Data 3). We then used the *lacZ* sequencing data from NGS and Sanger experiments in COSMIC format (Supplementary Fig. 4; Supplementary Data 4) to identify which of the normalized signatures were most closely associated with the mutation pattern of each agent. For this analysis, only sequenced single nucleotide substitutions were used resulting in a total of 3270 mutations (of these, 944 were from controls) that were used to query the COSMIC database. The distribution of the mutations among the 10 agents is shown in Supplementary Table 4.

Two complementary approaches were used. In the first approach, the mutation profile of each agent was compared individually against each of the SBS signatures in the COSMIC database. This initial analysis showed that mutational signatures in human cancers that have been associated with specific mutagenic exposures were enriched in the *lacZ* mutation profiles

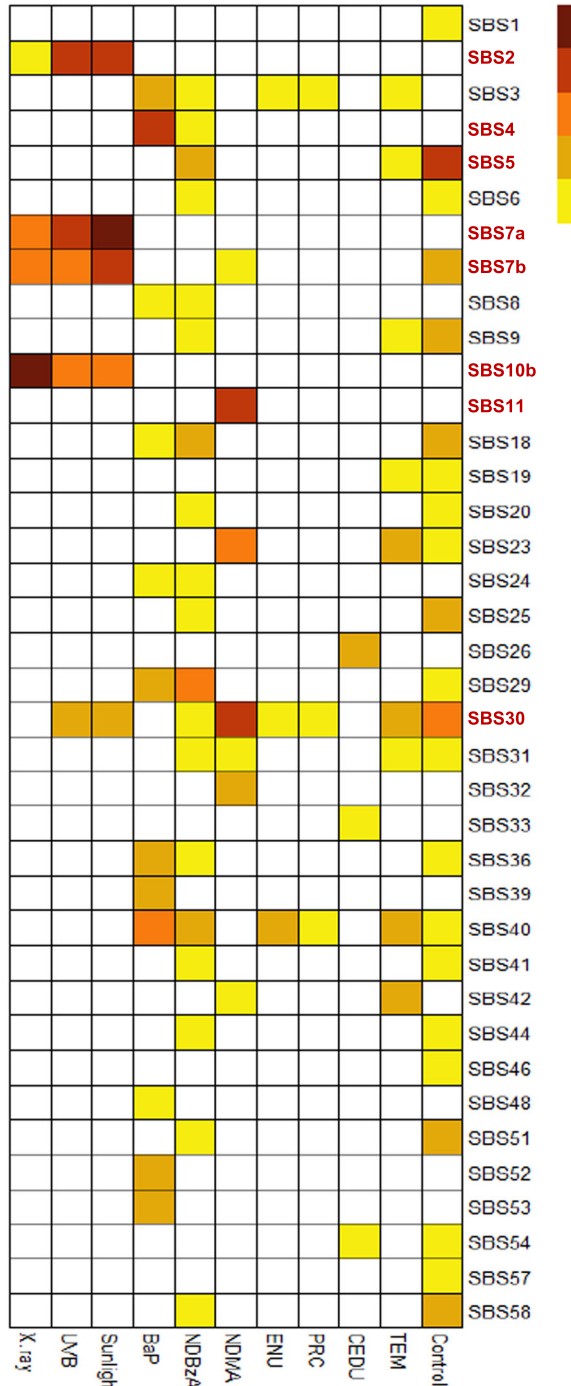

**Fig. 4 Heatmap of similarities between obtained mutational profiles of tested agents and COSMIC SBS signatures.** All comparisons that had a cosine similarity above 0.5 are shown. The eight SBS signatures that had a cosine similarity greater than 0.7 are indicated in bold on the right of the heatmap.

for the appropriate agent tested in this study (Fig. 4; Supplementary Data 5). For example, the UVB[29,31] and sunlight[30] mutation profiles had very strong correlations (cosine similarity = 0.86–0.94) with SBS 7a signature, which is observed in human skin cancers. Similarly, the BaP mutation profile showed a strong correlation with SBS 4 (cosine similarity = 0.81), which is observed in tobacco smoke-induced cancers, and SBS 40 (cosine similarity = 0.74), which has currently no known etiology[15]. In total, there were seven SBS signatures that had a cosine similarity

values greater than 0.8 with the mutation profiles generated from sequenced *lacZ* mutations induced by the various agents tested (Fig. 4).

In the second approach, we used two computational tools, deconstructSigs[53] and MutationalPatterns[54], to simultaneously query the entire COSMIC SBS database to investigate which of the signatures contributed to the observed mutation patterns. Prior to this analysis, we used control mutation data to generate an in vivo background signature (Fig. 5; Supplementary Data 4) to account for the fact that some mutations present in the exposure groups are also spontaneous in origin rather than specific to the mutagen tested. This is especially true for weak mutagens. The in vivo background signature is enriched primarily in C>T mutations, and to a lesser extent C>A mutations (Fig. 5). This was consistent among all tissues that contributed to the control signature (Supplementary Fig. 5). The two computational tools produced very similar results both in terms of suggested COSMIC signatures and their percent contribution (Supplementary Table 4). In addition, the reconstructed signatures (see, "Methods") had very high cosine similarity values (0.89–0.98) for six of the agents and high cosine similarity values (0.67–0.82) for four agents with the respective *lacZ*-generated mutation profiles (Supplementary Table 4). Finally, application of stringent filtering criteria (see "Methods") revealed the association of nine COSMIC SBS signatures with mutation data from the various exposure groups (Fig. 6).

The signatures produced by the three electromagnetic radiations (i.e., UVB, sunlight, and X-rays) appear to be broadly similar when visually assessing individual SBS signature heatmaps (Fig. 4). However, we found that different mutational processes may contribute to each signature. Specifically, we found that both SBS 2 and SBS 7a contributed to the mutation profile of sunlight and each explained ~25% of the data (Fig. 6). However, only SBS 2 was significantly associated with the UVB mutation profile and explained over 30% of its data. This may indicate that SBS 7a is a signature produced by the mutagenic components of sunlight other than UVB. In addition, the UVB mutation profile had a significant contribution (over 30%) from the control signature. Finally, the mutation pattern associated with X-rays, which induces large deletions rather than point mutations (56 indels ranging from 1 to 437 bp vs 35 SNVs[33]), was associated only with the SBS 10b signature (>40%) and the control signature (~20–25%).

For the bulky adduct group, the mutation pattern of BaP revealed mutational processes characteristic of SBS 4 (>30%) and SBS 39 (~25%) signatures. SBS 4 is most notably associated with tobacco-smoke-induced cancer[16], while SBS 39 is one of the new signatures that currently does not have a proposed etiology. As for the electromagnetic radiations, deconstructSigs and MutationalPatterns helped in identifying the signatures that are most likely to contribute to the mutation profile of BaP. In fact, even though SBS 29, 36, 40, 52, and 53 all have cosine similarity values >0.6 with the mutation profile of BaP (Fig. 4), they do not contribute significantly to its profile once SBS 4 is taken into account. No SBS signature was associated with the mutation profile of NDBzA and the control signature explained ~50% of the mutation profile of this agent.

Analysis of the alkylating agent exposure group revealed that SBS 11 and SBS 30 signatures were associated with N-nitrosodimethylamine (NDMA) mutation data[34] and explained 37 and 50% of the mutations, respectively. SBS 11 has previously been linked to exposures to the methylating agents temozolomide and N-methyl-N'-nitro-N-nitrosoguanidine[17,19]. SBS 30 is hypothesized to be associated with defects in base excision repair[15]. No SBS signatures were associated with the mutation profiles of ENU or PRC while the control signature explained ~35% and ~45–50% of the mutation profiles of these two chemicals, respectively.

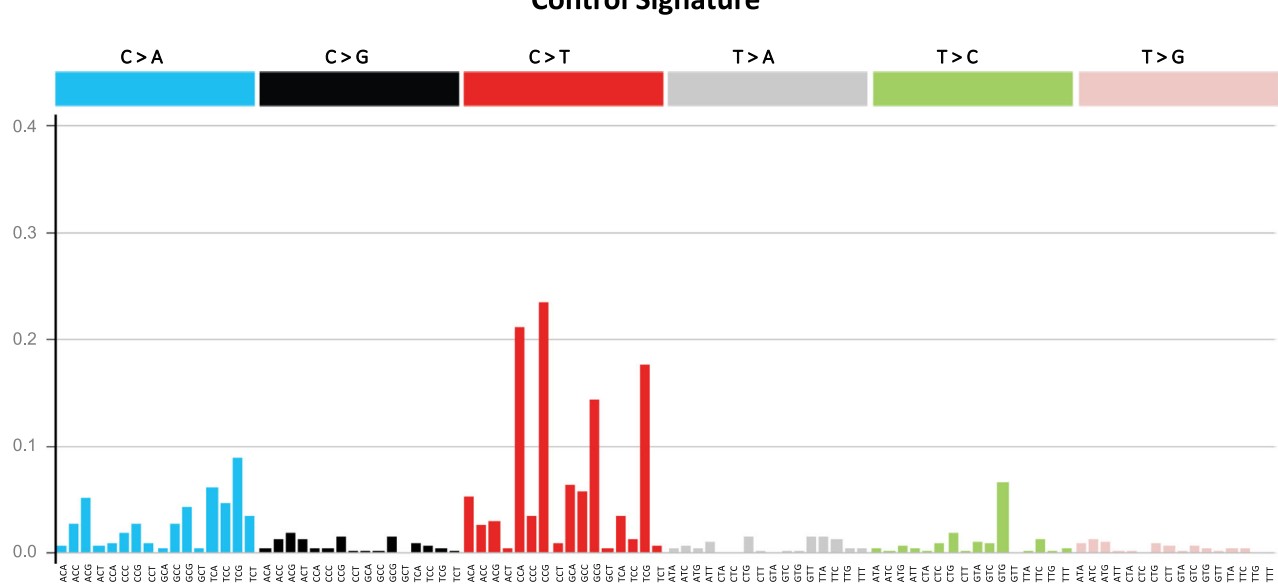

**Fig. 5 The lacZ control signature.** The control signature is based on empirical mutation data from control animals in NGS and Sanger studies.

| Signature | Electromagnetic radiation | | | | | | Bulky adducts | | | | Alkylating agents | | | | | | Base analog | | Clastogen | |
|---|---|---|---|---|---|---|---|---|---|---|---|---|---|---|---|---|---|---|---|---|
| | X rays (34) | | UVB (109) | | Sunlight (62) | | BaP (1165) | | NDBzA (76) | | NDMA (30) | | ENU (611) | | PRC (110) | | CEDU (14) | | TEM (115) | |
| | DS | MP | DS | MP | DS | MP | DS | MP | DS | MP | DS | MP | DS | MP | DS | MP | DS | MP | DS | MP |
| SBS 2 | | | 33 | 32 | 27 | 26 | | | | | | | | | | | | | | |
| SBS 4 | | | | | | | 36 | 31 | | | | | | | | | | | | |
| SBS 7a | | | | | 27 | 23 | | | | | | | | | | | | | | |
| SBS 10b | 49 | 43 | | | | | | | | | | | | | | | | | | |
| SBS 11 | | | | | | | | | | | 37 | 35 | | | | | | | | |
| SBS 26 | | | | | | | | | | | | | | | | | 80 | 43 | | |
| SBS 30 | | | | | | | | | | | 50 | 45 | | | | | | | | |
| SBS 39 | | | | | | | 26 | 24 | | | | | | | | | | | | |
| SBS 40 | | | | | | | | | | | | | | | | | | | 33 | 24 |
| Control | 25 | 20 | 33 | 34 | | | | | 53 | 53 | | | 35 | 36 | 42 | 41 | | | 20 | 20 |

**Fig. 6 The contribution of COSMIC signatures to the mutation profile of each agent.** The number below each agent indicates the number of unique mutants sequenced, while the number in each box represents the percent contribution of each signature to the mutation profile of each tested agent. Only those signatures that passed the criteria for inclusion (i.e., detected by both deconstructSigs and MutationalPatterns; at least 20% contribution by both methods; and cosine similarity >0.5 with the mutation profile) are shown. DS = deconstructSigs; MP = MutationalPatterns.

There were limited data available for nitrogenous base analogs. Data were only obtained from mice exposed to 5-(2-chloroethyl)-2-deoxyuridine (CEDU)[41], a uridine analog. This included only 14 characterized mutants from bone marrow, 13 of which were T>C mutations. CEDU represents the only agent for which deconstructSigs and MutationalPatterns produced different results. In fact, while they both detected SBS 26 as contributing to the mutation profile of CEDU, this signature explained 80% of the data according to deconstructSigs but only 43% of the data according to MutationalPatterns.

TEM had a SNV mutation pattern that was similar to controls (Fig. 2). Nevertheless, we found that SBS 40 contributed to ~30% of the TEM data, which was higher than the 20% that can be attributed to the control signature. There is currently no known etiology for the SBS 40 signature.

Finally, we used various strategies to assess the robustness of the association of the nine SBS signatures with the mutation profiles of the 10 agents tested. First, we analyzed the impact of increasing the minimum cosine value required to be considered for inclusion. This analysis showed that progressively increasing the cosine value from >0.5 to >0.8, resulted in the elimination of only three of the nine SBS signatures (Supplementary Table 5). Specifically, the associations between SBS 10b and X rays, SBS 2 and UVB, SBS 2 and SBS 7a with sunlight, SBS 4 and BaP, and SBS 11 and SBS 30 with NDMA were unaffected; conversely, the associations between SBS 39 and BaP, SBS 26 and CEDU, and SBS 40 and TEM were impacted by increasing stringency criteria and require further testing to be confirmed. Second, we randomly downsampled by 50% the number of mutations used as input for MutationalPatterns, and found that this does not change the

mutational signatures that are detected (Supplementary Fig. 6). The results demonstrate that even fewer mutations can be sufficient to detect a signal and suggest that the association between the mutation profiles and COSMIC signatures observed in this study is robust. Third, we explored random resampling of the mutation data to evaluate whether some of our results could be due to chance (Supplementary Fig. 7). This random reassignment of mutations to different trinucleotide patterns resulted mainly in the identification of flat signatures, that is, signatures that are not enriched in a specific type of base-pair alteration and do not have a proposed etiology (e.g., SBS 3, SBS 40) or are suspected sequencing artefacts (e.g., SBS 49). In addition, the cosine similarities between the reconstructions obtained for resampled data were very low compared to the cosine similarities between our original mutational profiles and their reconstructions (Supplementary Fig. 7).

## Discussion

We show that in vivo NGS–TGR data can be used to extract mutagenic mechanisms that may contribute to human cancers through the application of COSMIC signature analysis. We also show that such analyses are improved through the inclusion of a background mutational signature (i.e., control signature) that reflects spontaneous mutations resulting from endogenous processes. Analysis of induced mutations in mouse tissues following exposures to 10 mutagenic agents (two sequenced by NGS, six sequenced by the Sanger method, and two by both) revealed high concordance between the expected mutagenic mode of action and the relevant COSMIC signature. The data suggest that our approach may be used to: (i) test if TGR mutation patterns support hypotheses that COSMIC signatures are attributed to particular mutagenic exposures, and (ii) generate hypotheses about the mutagenic mechanisms underlying human cancers through identifying enriched COSMIC signatures in TGR mutation patterns.

A large portion of mutations collected from weak mutagens are spontaneous rather than chemically induced. Thus, we developed a background signature derived from our empirical control data that can be integrated with COSMIC signatures to reduce the noise in the mutation pattern of an agent that is attributable to spontaneous mutations. Indeed, we found that the control signature contributed to the mutation profile of six of the 10 agents investigated (Fig. 6). This does not mean that these agents operate through a common mechanism, but simply that the magnitude of the induced effect is insufficient to hide the contribution of spontaneous mutations. For example, in the case of TEM, which barely induces a two-fold increase in mutations (Supplementary Table 2), ~50% of the sequenced mutations are expected to be spontaneous in origin and not induced by TEM.

The in vivo control signature is a unique feature of our study, as there is currently no in vivo control signature reported in a recent study that generated chemical-specific signatures using a different approach[27]. As shown in Fig. 4, the background signature is most closely associated with SBS 5 (cosine similarity = 0.81), which is one of the two COSMIC signatures that contribute to the mutation burden in normal cells as function of age[22]. Our results show that C>T transitions are the most common spontaneous mutations in vivo (Fig. 5) and this was consistent among all tissues analyzed (Supplementary Fig. 5). C>T transitions at CpG sites are known hotspots of mutation due to spontaneous deamination of cytosine[49]. Previous work using bisulfite sequencing has shown that CpG sites in *lacZ* are heavily methylated, and CpG flanked by a 5′ pyrimidine were most likely to have C>T base substitutions[46]. This is supported by our control data: the most prevalent spontaneous mutations were C>T at

CCG, and, the third most prevalent were C>T mutations at TCG (Fig. 5). Thus, our background control signature is consistent with expectations.

An in vitro background signature was recently reported[28]; however, the correlation between the two control signatures is modest (cosine similarity = 0.56) because, at variance with our results, the in vitro control signature is enriched for C>A mutations. Spontaneous deamination of cytosine is also the most likely reason for C>A transversions and appears to be the most common spontaneous mutation in vitro[55]. This suggests that differences in oxidative and methylation status of cytosines between in vitro and in vivo may contribute to the different mutagenic outcome of cytosine deamination.

COSMIC signatures represent the repertoire of mutagenic mechanisms that have been identified by analyzing mutations observed in human cancers. The landscape of mutations in a fully-grown cancer can then be reconstructed as a combination of one or more COSMIC signatures using a variety of approaches[56]. Similarly, the mutation pattern of a mutagen can be thought of as the result of multiple mutagenic mechanisms, as it is unlikely that a mutagen induces only one type of DNA damage and that only one DNA repair pathway processes all induced lesions[57]. Thus, we applied deconstructSigs and MutationalPatterns to determine whether the mutation pattern of each agent could be explained in terms of COSMIC signatures. Application of the control signature (Fig. 5) and stringent statistical analysis identified nine SBS signatures that were associated with the *lacZ* SNVs induced by the investigated exposures. Two major outcomes from this analysis are: (1) mutation profiles for some of the tested agents were highly enriched for COSMIC signatures from cancers where the agents are known etiological factors (e.g., UV for skin cancer and BaP for tobacco-related cancers); and, (2) a few *lacZ* mutation profiles were associated with a variety of signatures of unknown aetiologies. This raises the question of whether the mutagenic mechanisms of these prototype agents are determinants of the signatures.

We identified SBS 2, SBS 7a, and SBS 10b signatures as important contributors to the mutagenic mechanisms of all three electromagnetic radiation agents investigated (i.e., X-ray, UVB, and sunlight). SBS 2 has been observed in ~14% of cancer samples and is present in 22 cancer types but is most often found in cervical and bladder cancers[14,17]. In this study, the signature was most strongly associated with UV skin exposure, representing 33–27% of mutations in exposed animals. Mechanistically, cytosine deamination is accelerated by UV exposure[58]; thus, it is possible that we observed SBS 2 in this study because of UV-dependent cytosine deamination. However, SBS 2 is not observed in skin cancers[17]. This suggests that mutations arising from UV-dependent cytosine deamination are not the primary drivers of the surveyed human skin cancers in the COSMIC database, and that other lesions (e.g., various types of photodimers) are the main contributors to the mutation catalog of UV-induced skin cancers. Another possible explanation is that with a small sample size of mutations, the high degree of similarity in the SBS 2 and SBS 7a signatures confounds this analysis. By this logic, some portion of the mutational signature identified as SBS 2 in our study may be the result of the mutational processes associated with SBS 7a, which is found in multiple cancer types but is most pronounced in skin cancers[14,17]. Indeed, the SBS 7a signature contributes to 27% of the mutations observed after sunlight exposure.

Activation of error-prone polymerases has been attributed to SBS 10b[14], a signature that is mostly found in colorectal and uterine cancers. In the present study, this signature was only associated with X-ray mutations (49%). X-ray mutations show a high proportion of C>T substitutions at the TCG motif

(Supplementary Fig. 4), which is characteristic of the *lacZ* normalized SBS 10b signature (Supplementary Fig. 3). It is possible that there is an ionizing radiation component to this signature. However, given previous work in this area, it is more likely that the association between SBS 10b and X-ray SNVs is a result of error-prone replication occurring in response to DNA damage.

The analysis of mutational signatures for the electromagnetic radiation agents provide support for the ability of the expanded repertoire of COSMIC signatures to exploit subtle differences in the mutation profiles to extract different mutational mechanisms. Using the previous version of the COSMIC database, all three radiation types had a comparable contribution from signature 7 (21–33%; Supplementary Table 6). However, there are now four SBS signatures (7a–7d) derived from the original signature 7 in the latest COSMIC database[15], and of these, only the SBS 7a signature contributes significantly to the mutation profile of sunlight.

Tobacco smoking is strongly associated with SBS 4, and this signature is commonly found in the lung tumors of smokers. SBS 4 is very similar to the mutation profile generated by BaP, a major mutagenic component in tobacco smoke[21], both in vivo[27] and in vitro[16,18,28]. In line with these findings, we found that SBS 4, which is enriched for C>A transversions at NCG sites, contributed the highest percentage (36%) to the mutation profile of BaP in our study. Two other signatures (SBS 29 and 36) with cosine similarity values >0.6 (Fig. 4) are also enriched in C>A transversions. However, both deconstructSig and MutationalPatterns showed that these two signatures do not contribute to the mutation profile of BaP once SBS 4 is taken into account. Thus, C>A mutations in the BaP profile are mainly driven by SBS 4. Interestingly, SBS 4 was the only signature that contributed to the mutation profile of BaP and accounted for 60% of the observed mutations when using the previous version of the COSMIC database (Supplementary Table 6). However, using version 3 of the COSMIC database[15], the contribution of SBS 4 declined while we identified a second signature that contributed to the BaP mutation profile. Specifically, we detected a significant contribution (~25%) of the SBS 39 signature, which is one of the new signatures and currently has no known etiology. The presence of SBS 39 in the mutation profile of BaP is driven by the occurrence of C>G transversions at NCT. These results suggest that SBS 39 may be associated with exposure to chemicals that induce bulky adducts at guanines.

The BaP mutation profile that we derived using our approach is consistent with previous work in vivo[27] and in vitro[28] that demonstrated the presence of SBS 4 after exposure to BaP. Indeed, the BaP mutation profile is consistent among the three studies (cosine similarities of 0.85 and 0.76 with the in vivo and in vitro profile, respectively). Remarkably, signatures SBS 24, which has been associated with aflatoxin adducts, and SBS 29, which has been associated with tobacco chewing, are strikingly similar to SBS 4 (Supplementary Fig. 3). However, only SBS 4 strongly correlates with the BaP mutation data. This demonstrates the robustness of the mutational signatures and the ability of TGR–NGS to correctly discriminate between similar signatures that have different aetiologies. It also emphasizes the importance of the flanking nucleotides to increasing the specificity of the signatures; this work demonstrates that 96-bp signatures provide superior mechanistic information to standard mutation pattern analysis.

NDMA was the only alkylating agent among those investigated that was associated with an established COSMIC signature. About 50% of the NDMA mutation profile was explained by the SBS 30 signature that has been associated with a deficiency in base excision repair. NDMA is known to induce mostly 06- and N7-methyl guanine adducts[34], thus, a role of base excision repair in the response to this chemical is expected. NDMA exposure was also enriched for SBS 11 (37%), inducing primarily C>T mutations at

CpC motifs (Supplementary Fig. 4). SBS 11 has been detected in melanomas and glioblastomas, and the mutation pattern of this signature has been attributed to alkylating agent exposures, such as temozolomide and N-methyl-N'-nitro-N-nitrosoguanidine[17,19]. These alkylating agents induce C>T mutations, mostly at CpC motifs, and mutations at this motif are the four most common in the SBS 11 signature. The TGR mutation data from our study are consistent with this expected mutation pattern.

The SBS 11 signature was not enriched within the mutation patterns of the two other alkylating agents (i.e., ENU or PRC) in our mutation database. This is expected because these compounds induce a very different mutation pattern, causing primarily T>A mutations. These differences demonstrate that SBS 11 is specific to a particular mechanism of alkylation (i.e., target sites for the alkylation events) and that there is currently no COSMIC signature for alkylating agents that target thymine. Further TGR–NGS analyses of alkylating agents may refine our understanding regarding which specific alkylating agents or defective alkyltransferases underlie the mechanisms associated with SBS 11.

The mutation profile obtained with ENU, demonstrating a slight preponderance of T>A mutations over T>C mutations, is consistent (cosine similarity = 0.90) with that obtained in the bone marrow of *gpt* delta mice[27], although the correlation is reduced when expanding the six possible base-pair alterations to the 96 possible mutation types (cosine similarity = 0.70). This is mostly due to a deficiency of T>C mutations at CTN motifs with respect to *gpt* delta mice. Nevertheless, the similarity with the ENU mutation profile from *gpt* delta mice is greater than that obtained in vitro with an induced pluripotent stem cell (iPSC) line (cosine similarity = 0.53) where the ENU signature is dominated by T>C mutations[28]. These authors speculate that the preponderance of T>C mutations after in vitro exposure to ENU is driven by the intrinsic characteristics of DNA repair processes in iPSCs.

The SBS 26 signature was enriched in the mutation profile of CEDU, a nitrogenous base analog; however, deconstructSigs and MutationalPatterns differed significantly in the percent amount of its contribution (80% vs. 43%, respectively). SBS 26 is one of the seven SBS signatures associated with defective mismatch repair, which is one of the major repair pathways that deals with base analogs[59]. Due to the limited number of mutations recovered in the CEDU study, the association between SBS 26 and CEDU should be further tested. Also, considering that CEDU is similar in structure to existing halogenated uracil analogs that serve as therapeutics (e.g., fluorouracil), attention should be given to these compounds as possible contributors to the SBS 26 signature and associated cancers.

Among the agents tested in this study, TEM is the only one that is more effective at inducing chromosomal structural aberrations than mutations. TEM is a trifunctional alkylating agent that induced a strong micronucleus response while eliciting a weak mutagenic response in the hematopoietic system[48]. Our analysis identified SBS 40 signature as a strong contributor (32%) to the mutation profile of TEM. SBS 40 is one of those signatures that is not dominated by any specific type of base-pair alteration and does not have a proposed etiology. Further studies are needed to confirm whether SBS 40 signature is an indicator of a clastogenic mode of action.

Overall, these results demonstrate that *lacZ* transgene sequence data may be used, in conjunction with established mutation signatures derived from COSMIC cancer data sets, to test the hypothesis that a given class of mutagenic agents is linked with specific human cancers. Moreover, COSMIC signature mining based on TGR mutation datasets can be used to generate new hypotheses regarding the mutagenic mechanisms associated with human cancers. This study presents a potential avenue through which mutation signature analysis can be applied to in vivo

experimental models, and the analyses employed to improve understanding of mode of action. The analyses can also generate hypotheses regarding the mutational mechanisms of uncharacterized chemicals.

There are a few limitations to our approach. While we demonstrate that characterization of mutational signatures shortly after exposure in a non-tumor target tissue produces meaningful information on potential human-relevant mechanisms of carcinogenesis, it is possible that the correlation between the mutation profiles for some of the tested agents and COSMIC signatures would have been even stronger had the analysis been conducted in tumor target tissues. Differences in metabolism, DNA repair, or polymerase enzymes preferentially used in cancer target tissues relative to non-cancer target tissues may have impacted the observed mutation signatures. Indeed, the mutation profiles of the electromagnetic radiations, which were generated in the principal tumor target tissue, had the highest cosine similarity values with the relevant COSMIC signatures (Fig. 4). Second, because the *lacZ* is transcriptionally inert in the MutaMouse model, our approach cannot be used to analyze strand bias in mutations due to transcription-coupled repair[60]. At the same time, this assures that any mutation induced in *lacZ* is recovered because it does not confer a fitness disadvantage to the cell carrying the mutation. Finally, we failed to identify COSMIC signatures contributing to the mutation profile of some of the agents tested. It is possible that analysis of a larger number of mutations would have identified a COSMIC signature for even these agents. However, this finding is consistent with the other two studies[27,28] that have attempted to decompose the mutation pattern of physical and chemical agents using the COSMIC database and had analyzed larger number of mutations. We suggest that: (1) COSMIC signatures do not yet capture all possible mutagenic mechanisms and are insufficient to appropriately decompose all mutagenic signatures; or (2) there is yet an insufficient number of cancers in the COSMIC database where these agents play a role in the carcinogenic process.

The in vivo TGR–NGS approach has comparable sensitivity to whole-genome approaches used for investigating the mutational landscape of environmental agents[18,19,26,28,61]. However, by avoiding the orders-of-magnitude higher cost of whole-genome sequencing, the in vivo TGR–NGS approach offers much higher-throughput for the testing of chemical mutagens. Overall, these results highlight that some mutational signatures may have large environmental components and contribute to the growing body of evidence that analyses of mutation spectra shortly after exposure has bearing on the carcinogenic mechanism and the mutational profile observed in fully developed cancers.

## Methods

**Animal treatment**. Male MutaMouse animals (8–15 weeks old; 6–8 per group) were exposed daily to either 100 mg/kg BaP, 5 mg/kg ENU, 25 mg/kg PRC or 2 mg/kg TEM by oral gavage for 28 days as per the Organisation for Economic Co-operation and Development (OECD) test guideline 488[62]. All doses were selected based on pilot studies conducted to identify the maximum tolerated dose as per TG 488 guidance. The BaP[8], PRC[47], and TEM[48] data are the same as presented in the respective reference. Matched controls received the solvent (olive oil or water) by oral gavage during the same period. Three days after the last daily exposure, mice were anaesthetized with isoflurane and euthanized via cervical dislocation. Bone marrow cells were isolated by flushing femurs with 1X phosphate-buffered saline. After brief centrifugation, the supernatant was discarded, and the pellet was flash-frozen in liquid nitrogen prior to storage at −80 °C. All animal procedures were carried out under conditions approved by the Health Canada Ottawa Animal Care Committee.

***lacZ* mutant quantification, collection, and sequencing**. The experimental protocol for enumerating *lacZ* mutants followed OECD guideline 488[62]. Briefly, bone marrow was thawed and digested overnight with gentle shaking at 37 °C in 5 mL of lysis buffer (10 mM Tris–HCl, pH 7.6, 10 mM ethylenediaminetetraacetic acid (EDTA), 100 mM NaCl, 1% sodium dodecyl sulfate (w/v), 1 mg/mL Proteinase K). High molecular weight genomic DNA was isolated using phenol/chloroform extraction as described previously[42,63]. The isolated DNA was dissolved in 100 μL

of TE buffer (10 mM Tris pH 7.6, 1 mM EDTA) and stored at 4 °C for several days before use. The phenyl-β-D-galactopyranoside (P-gal) positive selection assay[64] was used to identify *lacZ* mutants present in the DNA. Briefly, the λgt10lacZ construct present in the genomic DNA was isolated and packaged into phage particles using the Transpack™ lambda packaging system (Agilent, Mississauga, Ontario, Canada). The phages were then mixed with E. coli (*lacZ⁻*, *galE⁻*, *recA⁻*, *pAA119⁻* with galT and galK)[63] in order to transfect the cells with the *lacZ* construct. E. coli were then plated on a selective media containing 0.3% P-gal (w/v) and incubated overnight at 37 °C. Only E. coli receiving a mutant copy of *lacZ* where the gene function is disrupted can form plaques on the P-gal medium, because P-gal is toxic to galE⁻ strains with a functional *lacZ* gene product[1]. Packaged phage particles were concurrently plated on plates without P-gal (titer plates) to quantify the total plaque-forming units to be used as the denominator in the mutant frequency calculation.

After enumeration, plaques from each individual sample were collected and pooled together in microtubes containing autoclaved milliQ water (0.3 plaques/μL; mutants from 1 sample per tube). Mutant amplification and sequencing was done as described previously[8]. Briefly, the mutant pools were boiled for 5 mins and transferred to a PCR mastermix containing a final concentration of 1X Q5 reaction buffer, 200 μM dNTPs, 0.5 μM Forward primer (GGCTTTACACTTTATGCTTC), 0.5 μM Reverse Primer (ACATAATGGATTTCCTTACG), and 1U Q5 enzyme (New England BioLabs Ltd., Whitby, Ontario, Canada); the final volume of each PCR was 50 μL. To control for errors introduced during PCR, each mutant pool was amplified twice as two separate technical replicates. The following thermocycle program was used for amplification: 95 °C for 3 min; 30 cycles of 95 °C for 45 s, 50 °C for 1 min, 72 °C for 4 min; final extension at 72 °C for 7 min. PCR products were purified using the QIAquick PCR purification kit (Qiagen, Montreal, Quebec, Canada).

NGS libraries were built using the NEBNext® Fast DNA Library Prep Set for Ion Torrent™. Each technical replicate had a unique barcoded adaptor ligated to the *lacZ* DNA fragments allowing for many samples to be sequenced simultaneously (up to 96 libraries per NGS run). Sequencing was performed using the Ion Chef™ workflow and Ion Proton™ system with P1 chips. NGS reads were aligned to the *lacZ* gene using bowtie 2 (version 2.1.0) and read depths for every possible mutation were quantified using samtools (version 0.1.19). Mutations were called if, after background correction (determined by sequencing non-mutants), both technical replicates had mutation read depths above threshold values (equal to at least 1/number of plaques in pool)[8]. To further filter the data in this study, if the mutation read depths between two technical replicates varied by ≥50% then that mutation was removed from analysis. Clonally expanded mutants were only counted as one mutation.

**Published Sanger sequencing data**. Published data came from studies where *lacZ* transgene mutants were sequenced and the position and type of each mutation was reported (summarized in Supplementary Table 1). Mutants were characterized from MutaMouse or *LacZ* Plasmid mice[65]. Some studies reported the position of the mutation in the plasmid construct, while others reported the position in the coding sequence. For consistency, the positional information was adjusted to reflect the position of the mutation in the coding sequence of the *lacZ* gene. Furthermore, the reference sequence of *lacZ* used for NGS has four variations[38] relative to the E. coli *lacZ* coding sequence (Genbank: V00296.1)[66], including a 15 bp insertion into codon 8. Thus, mutation positions were also adjusted to reflect this where applicable (e.g., if *LacZ* Plasmid mice were used instead of MutaMouse). No mutations were detected at or next to the variant positions in the *LacZ* Plasmid motif. In contrast to NGS work, different tissues were used for these analyses (i.e., bone marrow, brain, colon, germ cells, kidney, liver, skin, spleen, and stomach). Tissue sources are noted in the results with the accompanying data.

**Signature analyses**. The workflow used to do signature analyses are available as an RShiny web-application (https://github.com/MarcBeal/HC-MSD/tree/master/lacZ_Mutations_COSMIC_Signatures and https://github.com/mattjmeier/lacZ_COSMIC). Mutations for control and exposed samples (see metadata in Supplementary Material) were imported into the R console[67] as VRanges using the package "VariantAnnotation"[68] with the *lacZ* coding sequence as the reference FASTA file. To determine which of the COSMIC mutation signatures best explained the observed *lacZ* mutant pattern, the COSMIC mutation signature weights, which are derived from human mutation data, were first normalized to *lacZ* trinucleotide frequencies. This was done using the ratio of trinucleotide frequencies in *lacZ* to the trinucleotide frequencies in the human genome (Fig. 3; the normalized signatures are shown in Supplementary Fig. 3 and the raw numbers in Supplementary Material). Analysis was done this way (as opposed to converting *lacZ* mutation data themselves to human trinucleotide frequencies) because the COSMIC signatures are based on a much larger database, and therefore, represent a more robust signal with less variance. Following normalization, each of the 96 trinucleotide substitutions within each signature were represented as the relative frequency (i.e., all values in a signature sum to 1) by dividing each normalized value by the sum of all values for that signature. The trinucleotide mutation context (i.e., the nucleotide immediately upstream and downstream of the mutation) was obtained with the "mutationContext" function and converted to a motif matrix using the "motifMatrix" function (both in the "SomaticSignatures" package[69]). The motif matrix was then transposed to obtain the required format, and finally

decomposed into the constituent *lacZ*-normalized signatures using the "which-Signatures" function from "deconstructSigs"[53] or the 'fit_to_signatures' function in MutationalPatterns[54]. The contribution of each identified signature to the mutation data was reported as a fraction. If the sum of each signature did not account for 100% of the mutation data, then the remainder was reported as the "residual".

In order to account for spontaneous mutations often present alongside induced mutations, which is especially true for weak mutagens, we generated a signature for the spontaneous mutation background using the mutations observed in control animals. This included all control mutations characterized by NGS and Sanger sequencing. However, spontaneous SNVs characterized by Sanger sequencing were heavily biased towards positions 1072, 1090, 1187, 1627, and 2374. Therefore, Sanger sequencing data at these 5 positions were not used for deriving the control mutation signature. Signatures were plotted using ggplot2[70].

"Signature reconstruction" was then used to determine how well the combination of normalized signatures, identified using the signature fitting methods described above for deconstructSig and MutationalPatterns, explain the mutation data from the respective exposure groups. For example, if signatures 3 and 4 contributed 40 and 60% to the mutation profile of a compound, respectively, then the motif matrices for signatures 3 and 4 were multiplied by 0.4 and 0.6, respectively, and summed together. The reconstructed signature was then compared against the motif matrices of the compound using cosine similarity correlation.

Lastly, the contribution of individual signatures was further validated using cosine similarity. Specifically, each signature was compared against the respective 96-base context mutation spectra from which the signature was identified. In the final results, COSMIC signatures were reported as contributing to the mutation profile of an agent only if: (i) they were identified by both deconstructSigs and MutationalPatterns; (ii) their contribution was at least 20% by both approaches; and (iii) the cosine similarity with the mutation profile was greater than 0.5.

**Statistics and reproducibility**. Statistical analyses were done using the R programming language[67] using the animal as the experimental unit. Mutant frequencies were compared between exposure groups and controls using generalized estimating equations assuming a Poisson distribution for the error, as done previously[8], using the geepack library[71] with outliers (1 in control, 1 in TEM) removed. Bonferroni correction for multiple comparisons was used to adjust the threshold of significance. Mutation spectra of the chemical exposure groups were compared against controls using mutation proportions. The standard error for the mutation spectra was determined using error propagation. Significant differences in mutation spectra between chemically induced mutants and spontaneous control mutants were determined using Fisher's exact tests with Bonferroni correction for multiple comparisons (i.e., across different chemical groups). To compare whole mutation spectra between control and exposed groups, Fisher's exact tests were performed with Monte Carlo simulation with 10,000 replicates. Fisher's exact tests were also performed on $2 \times 2$ sub-tables for each mutation type.

**Reporting summary**. Further information on research design is available in the Nature Research Reporting Summary linked to this article.

## Data availability

Sequenced mutations generated in-house for BaP, ENU, PRC, and TEM are available on the Sequence Read Archive under BioProject accession number PRIJNA 640660. Sequenced mutants from all other agents were obtained from the published literature.

## Code availability

The workflow used to do signature analyses are available as an RShiny web-application (https://github.com/MarcBeal/HC-MSD/tree/master/lacZ_Mutations_COSMIC_Signatures and https://github.com/mattjmeier/lacZ_COSMIC). Others are publicly available open source R libraries (eg decontstructSigs).

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

## Acknowledgements

We would like to thank Angela Dykes, Lynda Soper, and John Gingerich for their technical contributions to this research. We are grateful for the advice provided by Dr. Ludmil Alexandrov and Andrew Williams. Funding for this research was provided for by Health Canada's Chemicals Management Plan and Genomics Research and Development Initiative.

## Author contributions

M.A.B., C.M., M.J.M., and J.O.B. conducted the MutaMouse animal studies and collected samples. M.A.B and M.J.M. sequenced plaques. M.A.B., M.J.M., and D.L. conducted the COSMIC analyses. C.Y. and F.M. secured funding for the study and were responsible for study conception and design. All authors contributed to data analysis, interpretation, paper writing and approved the final version of the paper.

## Competing interests

The authors declare no competing interests.
