## [Peer Review File · Communications Biology]

Reviewers' comments:

Reviewer #1 (Remarks to the Author):

COMMSBIO-19-1762

This manuscript provides compelling evidence that the chemically induced mutations in the MutaMouse LacZ when coupled with either next generation sequencing or Sanger Sequencing can be used to predict the mutational signatures identified by whole genome sequencing associated with environmental exposures to carcinogens. The advantage of the new method is its cost reduction, and how the methods can be used to explain mechanisms of mutagenesis. A unique feature of the assay that it also generates a mutational signature that results from presumptive exposure to endogenous carcinogens and can be used to enrich the signatures that are chemically dependent. As the method detects mutations that occur in vivo it also has the potential to compensate for biological selection and the recruitment of error prone translesion bypass polymerases which could be emphasized. A number of issues require attention.

1. The authors need to correct their terminology. They frequently refer to mutational spectrum when no spectrum was measured. Mutational spectrum refers to the incidence of mutation by codon e.g. mutational spectrum of p53. Mutation pattern refers to SNV e.g. transitions and transversions, and the sequence context refers to mutational signature. In their study they measure both patterns and signature but do not measure a spectrum. Please eliminate the term "mutational spectrum."

2. The authors should provide a brief description of why they measure bone marrow cell LacZ mutations to study mutation derived from BaP when this is a lung carcinogen. The authors need to address whether there will be tissue specific mutational outcomes that may occur due to differences in DNA-adduct repair (NER and BER) and translesional synthesis by error prone translesional bypass DNA polymerases.

3. The daily exposure to 100 mg/Kg BaP seems excessive and does not reflect human exposure. Just because the SBS4 signature observed in lung cancer due to smoking is replicated does not necessarily support causality due to differences in exposure level. This draw back should be discussed in the Discussion.

4. The authors do not discuss the strand bias that exists in chemically induced mutagenesis. There should be bias to the untranscribed strand. Please discuss.

5. The C>A SNVs observed with BaP while consistent with bulky DNA exocyclic adducts formed with N2-guanine they can arise by other mechanisms. For example, the same mutations can arise from 8-oxo-G a product of redox cycling BaP-quinones and by N7-guanine depurinating adducts. The authors could perhaps eliminate these alternative mechanisms by examining other SBS in the COSMIC data base.

Minor:

Line 273, the consequences of cytosine deamination C>T versus C>A does not depend on physiological context but on methylation status of C. Please correct.

Reviewer #3 (Remarks to the Author):

In this manuscript, Beal et al. present the applicability of transgenic (TG) rodent models to elucidate chemical mutagenicity signatures. The authors use the MutaMouse model, which is widely used in toxicological research studies to analyze mutations in response to chemical exposure. First, they identified 2,751 lacZ mutations in the bone marrow of the animals exposed to four chemical mutagens (i.e., Benzo[a]pyrene, N-ethyl-N-nitrosourea, procarbazine, and triethylenemelamine) using next-generation sequencing (NGS) or Sanger sequencing. In addition, the authors collected the mutation data in response to other eight mutagens from published literature and analyzed 96-type trinucleotide signatures. As the frequency of each trinucleotide is different between the human genome and the lacZ gene in MutaMouse, they calculated the lacZ-normalized set of the 49 COSMIC single-base substitution (SBS) signatures. The authors analyzed the associations between the signatures obtained from TG models and those from human cancers based on the Pearson's coefficient value and employed the deconstructSigs package to break-down signatures of a single mutagen into several components. The authors represented chemical mutagen signatures as the mixture of COSMIC signatures and, as a result, found some association between chemical signatures and human signatures, such as between B[a]P and SBS 4 or 39 signatures. Although I agree that the mutation analysis using the TG rodent model may be used to determine chemical mutagenicity signatures as claimed by the authors, I found some deficiencies in the approach of data analysis or interpretation of the results. I recommend that this manuscript should not proceed to the further reviewing process and be returned to the authors. I suggest that the authors re-submit after re-constructing the logical architecture of the manuscript.

Major issues

1. I disagree with the application of the deconstructSigs package to decompose signatures by "single" chemical mutagens. I recommend that the results or discussion related to the data obtained using deconstructSigs (Fig. 1 Steps 7-9, and Fig. 6) be removed from the manuscript due to the following reasons:

First, the manuscript fails to clearly explain the rationality or necessity to decompose signatures of "pure single chemicals" into several components. The deconstructSigs program has been originally developed to decompose signatures observed in human cancers, and its application for decomposition of signatures from human cancers is reasonably understandable because cancer patients are thought to be exposed to "multiple mutagens." However, the extension of this model to the decomposition of mutations by "pure single chemicals" is not straightforward. I think that the authors should explain the rationality underlying this analysis. For instance, the signature by NDMA was decomposed into signature 11 and 30. However, no sufficient explanation was provided related to the reason why the signatures by a single pure chemical were decomposed into two separate signatures. The authors should explain the hypothesis underlying this mechanism to allow interpretation of these results with the support of biological data here and for other mutagens evaluated.

The authors should demonstrate why the current COSMIC DB data were sufficient to appropriately decompose chemical mutagen signatures. As the results of deconstructSigs may change based on the data included in the signature database, I believe that the reliability and reproducibility of the decomposition results should be carefully judged. The application of deconstructSigs for the analysis of human cancers seems appropriate because the sources of the DB are human cancer data. However, this may not imply that it is also sufficient for the decomposition of mutagens evaluated in this study. I recommend that the similarity or association between signatures should be discussed based on the value of cosine similarity, which is unaffected by the content of DB.

2. I think the number of mutations analyzed is too small to elucidate the pattern of mutagen signatures with substantial reliability, especially for ENU, PRC, TEM and the mutagens listed in Supplementary Table S1. It may be easily anticipated that more or less than 100 mutations for the analysis of the data of 96 dimensions is insufficient. The pattern with the highest similarity to the pattern of ENU, PRC, TEM was that of the control, as indicated in Fig. 6. I think these data show that

the number of mutations are insufficient to elucidate chemical-specific signatures. Although there has been no clear criteria about the number of mutations, at least 500 to 1000 mutations should be analyzed. I recommend that the authors increase the number of mutations analyzed concerning these mutagens. As the fold induction in MutaMouse assay was relatively smaller for these agents as compared to that for B[a]P, the number of mutations may be increased by optimizing dose settings. The authors should indicate the range of variation between samples based on standard deviation (S.D).

Minor issues

1. The order of the substitution pattern on the horizontal axis of the mutation spectrum graph should be rearranged to that of signature graph.
2. I recommend that the similarity of signatures should be discussed based on cosine similarity, not Pearson's coefficient, in the manner similar to that employed in human cancer research.
3. In Discussion, the information on "background COSMIC signature" should be deleted. The concept of "background signature" in human cancer is scientifically inappropriate.

COMMSBIO-19-1762

Beal et al – Sequencing chemically induced mutations in the MutaMouse lacZ reporter gene identifies human cancer mutational signatures.

Responses to reviewers

We thank the reviewers for the constructive and helpful comments that they have provided. We believe that the revised manuscript is improved because of their input. Below we provide our responses to the reviewers' comments. The line numbers in our responses refer to the line numbers in the version of the revised manuscript with track changes.

Associate Editor's comments,

- i) Eliminate the possibility of other mechanisms for the DNA adducts by looking at SBS in the COSMIC database (point 5 by reviewer 1)
- ii) Clarify the rationale for using deconstructSigs (usually used for cancers which usually arise from multiple mutagens) and here the authors focus on single mutagens (reviewer 2).
- iii) Discuss and demonstrate that the COSMIC database is sufficient to decompose mutational signatures (reviewer 2).
- iv) Address the point about the number of mutations included in the analysis (reviewer 2)
- v) Clarify some points raised by the reviewers about the rationale and/or in the discussion as well as other address textual/figure suggestions (reviewer 1 and 2).

We are pleased that the reviewers think that our work is of considerable potential interest and we have carefully considered their comments. As requested, we have addressed all questions raised by the reviewers, including the 5 points summarized above by the Associate Editor and have revised the manuscript accordingly.

We note that on a couple of occasions, we have disagreed with the reviewers' comments and have provided a detailed explanation of why we disagreed.

Reviewers' comments:

Reviewer #1 (Remarks to the Author):

COMMSBIO-19-1762

This manuscript provides compelling evidence that the chemically induced mutations in the MutaMouse LacZ when coupled with either next generation sequencing or Sanger Sequencing can be used to predict the mutational signatures identified by whole genome sequencing

associated with environmental exposures to carcinogens. The advantage of the new method is its cost reduction, and how the methods can be used to explain mechanisms of mutagenesis. A unique feature of the assay that it also generates a mutational signature that results from presumptive exposure to endogenous carcinogens and can be used to enrich the signatures that are chemically dependent. As the method detects mutations that occur in vivo it also has the potential to compensate for biological selection and the recruitment of error prone translesion bypass polymerases which could be emphasized. A number of issues require attention.

1. The authors need to correct their terminology. They frequently refer to mutational spectrum when no spectrum was measured. Mutational spectrum refers to the incidence of mutation by codon e.g. mutational spectrum of p53. Mutation pattern refers to SNV e.g. transitions and transversions, and the sequence context refers to mutational signature. In their study they measure both patterns and signature but do not measure a spectrum. Please eliminate the term “mutational spectrum.”

We thank the reviewer for pointing that out and we have corrected the terminology throughout the paper by replacing “mutation spectrum” with “mutation pattern”. See for example, line 25 on page 2, and lines 51 on page 3 of the revised manuscript.

2. The authors should provide a brief description of why they measure bone marrow cell LacZ mutations to study mutation derived from BaP when this is a lung carcinogen. The authors need to address whether there will be tissue specific mutational outcomes that may occur due to differences in DNA-adduct repair (NER and BER) and translesional synthesis by error prone translesional bypass DNA polymerases.

We have selected the bone marrow as the tissue to study because it is one of the most commonly used tissues for mutagenicity assessment for regulatory purposes. We have added text to clarify this point (see lines 107-108 on page 5 of the revised manuscript). We note that in animal studies, BaP has been shown to induce cancer in a variety of tissues, including lung as the reviewer indicates, but also in forestomach, liver, oesophagus, tongue, and in hematopoietic tissues [IARC Monograph Volume 100F, 2012]. Thus, for BaP, bone marrow is one of the cancer target tissues.

We acknowledge that not all agents analyzed in our study were analyzed in the respective cancer target tissue. However, as demonstrated by the studies of Kucab et al (2019), which used an in vitro system with a human induced pluripotent stem cell (iPSC) line, and Matsumura et al (2019), which also used the bone marrow, analyses of mutations in a non-tumour target tissue shortly following exposure are informative for identifying mutational signatures that are observed in fully developed cancers where the tested agent is a known etiological factor. As discussed in the original manuscript, now lines 94-98 on page 5 of the revised manuscript (underlined text indicates added text): “Recent work in vivo²⁷ and in vitro²⁸ has shown that chemical-specific signatures detected shortly after exposure in non-tumour target tissues match signatures seen in human cancers. Thus, characterization of short-term mutational signatures in non-tumour tissues is a valuable approach to elucidate human-relevant mechanisms of carcinogenesis.”

The reviewer is also correct that tissue specific difference in DNA-adduct repair may play a role in the differential response of tissues. As noted in our discussion, now lines 390-400 on

pages 18 of the revised manuscript, the mutation profile of ENU that we generated in the bone marrow is closer to the mutation profile of ENU obtained in bone marrow by Matsumura et al, than that obtained by Kucab et al in the iPSC line, and speculated that this may be due in part to different repair capacities of the two cell types. In addition, we have added further text to a paragraph on study limitations indicating that differences in chemical metabolism, DNA repair or polymerase enzymes may play a role in the differences between signatures in cancer-target tissues versus the tissues used in our study (see lines 432-434 on page 19 of the revised manuscript).

3. The daily exposure to 100 mg/Kg BaP seems excessive and does not reflect human exposure. Just because the SBS4 signature observed in lung cancer due to smoking is replicated does not necessarily support causality due to differences in exposure level. This drawback should be discussed in the Discussion.

The reviewer is correct, and we have revised text in the discussion to remove any suggestion of causality (see lines 342-345 on page 16 of the revised manuscript).

4. The authors do not discuss the strand bias that exists in chemically induced mutagenesis. There should be bias to the untranscribed strand. Please discuss.

As we had indicated in the original manuscript (now lines 47-50 on page 3 of the revised manuscript), the *lacZ* gene is transcriptionally inert in the MutaMouse, and thus, cannot be used to analyze strand bias in mutations due to transcription coupled repair. We have added a paragraph to address the limitations of our approach including the inability to analyze mutation strand bias (see lines 436-439 on page 20 of the revised manuscript).

5. The C>A SNVs observed with BaP while consistent with bulky DNA exocyclic adducts formed with N2-guanine they can arise by other mechanisms. For example, the same mutations can arise from 8-oxo-G a product of redox cycling BaP-quinones and by N7-guanine depurinating adducts. The authors could perhaps eliminate these alternative mechanisms by examining other SBS in the COSMIC data base.

The reviewer is correct that BaP-induced mutations, or those of any other agent for that matter, can arise through a multitude of DNA damage and repair mechanisms. Indeed, when the mutation pattern of BaP is compared with each SBS signature individually, there are several SBS signatures with high cosine similarities. As indicated in Figure 4, SBS 4 has a cosine similarity greater than 0.8 with the mutation pattern of BaP, SBS 40 has a cosine similarity greater than 0.7, and several others, ie, SBS 3, 29, 36, 39, 52 and 53 have a cosine similarity greater than 0.6. Ignoring SBS 52 and 53, which are considered potential sequencing artefacts, C>A mutations are enriched in the *lacZ*-normalized SBS 4, 29 and 36 signatures. The most common mutation in these three signatures occur at CCG, GCG, and GCT sites, respectively. When we deconstructed the mutation pattern of BaP by querying the entire COSMIC database (Figure 6), we found that the mutation pattern of BaP is best explained by having a major contribution from SBS 4 and SBS 39, while none of the other SBS signatures passed our criteria for inclusion, that is, having a contribution higher than the highest residual and a cosine similarity greater than 0.5 (Figure 6). Thus, C>A mutations in the BaP profile are mainly driven by SBS 4. The presence of SBS 39 in the mutation profile of BaP is driven by the occurrence of C>G mutations at NCT and

represent another mechanism of mutation induction. We added a few sentences about this in lines 228-232 on page 11 and lines 345-349 on page 16 of the revised manuscript.

Minor:

Line 273, the consequences of cytosine deamination C>T versus C>A does not depend on physiological context but on methylation status of C. Please correct.

We thank the reviewer for the comment. We realized that our original text was not clear and we have revised it (see lines 291-293 on page 14 of the revised manuscript).

Reviewer #3 (Remarks to the Author):

In this manuscript, Beal et al. present the applicability of transgenic (TG) rodent models to elucidate chemical mutagenicity signatures. The authors use the MutaMouse model, which is widely used in toxicological research studies to analyze mutations in response to chemical exposure. First, they identified 2,751 lacZ mutations in the bone marrow of the animals exposed to four chemical mutagens (i.e., Benzo[a]pyrene, N-ethyl-N-nitrosourea, procarbazine, and triethylenemelamine) using next-generation sequencing (NGS) or Sanger sequencing. In addition, the authors collected the mutation data in response to other eight mutagens from published literature and analyzed 96-type trinucleotide signatures. As the frequency of each trinucleotide is different between the human genome and the lacZ gene in MutaMouse, they calculated the lacZ-normalized set of the 49 COSMIC single-base substitution (SBS) signatures. The authors analyzed the associations between the signatures obtained from TG models and those from human cancers based on the Pearson's coefficient value and employed the deconstructSigs package to break-down signatures of a single mutagen into several components. The authors represented chemical mutagen signatures as the mixture of COSMIC signatures and, as a result, found some association between chemical signatures and human signatures, such as between B[a]P and SBS 4 or 39 signatures. Although I agree that the mutation analysis using the TG rodent model may be used to determine chemical mutagenicity signatures as claimed by the authors, I found some deficiencies in the approach of data analysis or interpretation of the results. I recommend that this manuscript should not proceed to the further reviewing process and be returned to the authors. I suggest that the authors re-submit after re-constructing the logical architecture of the manuscript.

Major issues

1. I disagree with the application of the deconstructSigs package to decompose signatures by "single" chemical mutagens. I recommend that the results or discussion related to the data obtained using deconstructSigs (Fig. 1 Steps 7-9, and Fig. 6) be removed from the manuscript due to the following reasons:

First, the manuscript fails to clearly explain the rationality or necessity to decompose signatures of "pure single chemicals" into several components. The deconstructSigs program has been originally developed to decompose signatures observed in human cancers, and its application for decomposition of signatures from human cancers is reasonably understandable because cancer patients are thought to be exposed to "multiple mutagens." However, the extension of this model to the decomposition of mutations by "pure single chemicals" is not straightforward. I think that the authors should explain the rationality underlying this analysis. For instance, the signature by

NDMA was decomposed into signature 11 and 30. However, no sufficient explanation was provided related to the reason why the signatures by a single pure chemical were decomposed into two separate signatures. The authors should explain the hypothesis underlying this mechanism to allow interpretation of these results with the support of biological data here and for other mutagens evaluated.

We respectfully disagree with the reviewer. It is an oversimplification to assume that one chemical means one mutagenic mechanism because that would imply that a mutagen induces only one type of damage. This is rarely the case and any mutagenic agent induces different types of DNA damage, both directly and indirectly, and thus, multiple repair mechanisms are operating. Even in the case of strong mutagens such as BaP and ENU, point mutations are only one of the genotoxic outcomes. For example, both chemicals are strong inducers of micronuclei, which implies the generation of DNA double strand breaks. Thus, it can be expected that the mutation pattern of a chemical is a composite of multiple mutagenic mechanisms.

As the reviewer indicates, the COSMIC signatures represent the repertoire of mutagenic mechanisms that have been identified by analyzing the mutations observed in human cancers. The goal of our study was to determine whether the mutation pattern of each agent could be explained in terms of COSMIC signatures. For this, we applied two complementary approaches: the cosine similarity analyses in which the mutation profile of an agent was compared against each individual COSMIC signature and the deconstructSigs approach which attempted to decompose the mutation profile of each agent into the underlying COSMIC signatures.

The results of the two approaches were highly comparable and supported each other. For example, as noted by the reviewer, deconstructSig showed that the mutation profile of NDMA is a combination of SBS 30 (50%) and SBS 11 (37%). This is in line with the results of the cosine similarity analyses (Figure 4) which showed that SBS 30 and SBS 11 have cosine values of 0.85 and 0.83 with the mutation profile of NDMA, respectively. However, two examples discussed in the paper show the advantage of applying the deconstructSigs approach to the analysis of the mutation profiles of the agents in our study.

The mutation patterns of UVB and Sunlight have strong cosine similarity with SBS2 and SBS7 (Figure 4). However, when each mutation pattern is deconstructed, a difference becomes apparent. SBS2 contributes to the mutation profile of each agent. Conversely, when the contribution of SBS2 is taken into account, SBS7 becomes irrelevant for UVB, while it still plays an important role in the mutation pattern of sunlight (Figure 6). This may indicate that SBS2 is a signature of UVB, while SBS7 is a signature of other mutagenic components of sunlight. This was already discussed in the original manuscript (see lines 216-220 on page 10 of the revised manuscript). Additional text was also added in lines 221-222 on page 11 of the revised manuscript.

The second example is BaP and it has been discussed in our response to comment No 5 from reviewer 1.

We agree with the reviewer that we had not properly explained our hypothesis and rationale for using deconstructSig in our study. Thus, we have revised and added text to address this comment in several places in the revised manuscript (see lines 186-188 on page 9; lines 199- 202 on page 10; and lines 294-301 on page 14).

The authors should demonstrate why the current COSMIC DB data were sufficient to appropriately decompose chemical mutagen signatures. As the results of deconstructSigs may change based on the data included in the signature database, I believe that the reliability and reproducibility of the decomposition results should be carefully judged. The application of deconstructSigs for the analysis of human cancers seems appropriate because the sources of the DB are human cancer data. However, this may not imply that it is also sufficient for the decomposition of mutagens evaluated in this study. I recommend that the similarity or association between signatures should be discussed based on the value of cosine similarity, which is unaffected by the content of DB.

The reviewer is correct that the results of the deconstructSigs may change based on the data included in the database. For example, as indicated in lines 349-355 on page 16 of the revised manuscript, Signature 4 was sufficient to explain the mutation pattern of BaP when using the previous version of the COSMIC database, while with the latest update, the mutation pattern of BaP is best explained by a combination of SBS 4 and SBS 39, one of the COSMIC signatures that did not exist in the previous set of signatures. It is possible that if the repertoire of COSMIC signatures changes in the future, the decomposition of the mutation patterns of the agents will change as well but this is not a limitation of the deconstructSigs approach. Furthermore, as already noted in our previous response, there is good concordance between the COSMIC signatures that are identified using the deconstructSig approach and those with cosine similarity values above 0.7 suggesting that both approaches identify the same COSMIC signatures.

A common finding of the three studies that have attempted to decompose the mutation pattern of physical and chemical agents using the COSMIC database, ie, Kucab et al 2019, Matsumura et al 2019 and our study, is that it is not always possible to describe the mutation pattern of an agent in terms of COSMIC signatures. There are two possibilities: (1) the COSMIC signatures do not yet capture all possible mutagenic mechanisms and, as the reviewer is suggesting, they are insufficient to appropriately decompose all mutagenic signatures; or (2) there is yet an insufficient number of cancers in the COSMIC database where these agents play a role in the carcinogenic process. We have expanded on what we had already discussed in the original version of the manuscript to better explain this point (see lines 440-488 on page 20 of the revised manuscript).

2. I think the number of mutations analyzed is too small to elucidate the pattern of mutagen signatures with substantial reliability, especially for ENU, PRC, TEM and the mutagens listed in Supplementary Table S1. It may be easily anticipated that more or less than 100 mutations for the analysis of the data of 96 dimensions is insufficient. The pattern with the highest similarity to the pattern of ENU, PRC, TEM was that of the control, as indicated in Fig. 6. I think these data show that the number of mutations are insufficient to elucidate chemical-specific signatures. Although there has been no clear criteria about the number of mutations, at least 500 to 1000

mutations should be analyzed. I recommend that the authors increase the number of mutations analyzed concerning these mutagens.

Obviously, we agree that the more mutations are available the higher is the confidence in the results. However, we disagree that it is always necessary to analyze at least 500 mutations as the reviewer is suggesting. Below we provide several lines of evidence to support that the number of mutations that we have analyzed were sufficient to detect chemical-specific COSMIC signatures, if they existed.

- (a) As the reviewer points out, COSMIC SBS signatures have 96 possible mutation types. However, for each signature, only a subset of these 96 possibilities drive the signature. For example, SBS 4 is characterized almost exclusively by C>A mutations. Even a data set of a few dozen mutations would be sufficient to exclude SBS 4 if the majority of mutations were not C>A.**
- (b) For BaP and ENU, we have data from both our TGR-NGS approach and by Sanger sequencing. For both chemicals, the number of mutations is significantly higher for the former than for the latter (specifically, Sanger mutations were about 33% and 5% of the total mutations for ENU and BaP, respectively). Despite that, when TGR-NGS and Sanger generated mutations are analyzed separately, they produce the same result. We have added a new panel in Supplementary Figure 5B to show that the two sets of data for both chemicals cluster together and text to point this out (see lines 452-4555 on page 20 of the revised manuscript).**
- (c) The fact that we could not identify a COSMIC signature associated with ENU is in line with the results of Kucab et al (2019) and Matsumura et al (2019). These two studies had larger number of mutations than ours and yet they also failed to detect a COSMIC signature associated with ENU exposure. As mentioned in our previous response, we believe that this is more an indication that exposure to ENU, a chemical that is seldom found in the environment, does not contribute greatly to the pattern of mutations observed in human cancers. In addition, the largest study to date (Kucab et al) failed to identify COSMIC signatures for almost 50% of the chemicals tested despite having a large number of mutations for each of the agents. Thus, it is not surprising that we were not able to identify COSMIC signatures for several of the agents that we have analyzed.**
- (d) Although the number of mutations we had for the electromagnetic agents were fewer than those for ENU, PCR and TEM, we identified the expected signatures.**

Thus, we believe that our failure to identify COSMIC signatures after exposure to several of the agents investigated in our study is primarily due to the lack of contribution of these agents to the pattern on mutations detected in human cancers. However, the possibility that other factors, including differences between tumour target tissue and our non-tumour target tissues (i.e., see response No. 2 to reviewer #1) and the number of mutations analyzed, cannot be excluded. We have added text to discuss these possibilities (see lines 427-448 on pages 19-20 of the revised manuscript).

As the fold induction in MutaMouse assay was relatively smaller for these agents as compared to that for B[a]P, the number of mutations may be increased by optimizing dose settings. The

authors should indicate the range of variation between samples based on standard deviation (S.D).

The four agents that we analyzed with our TGR-NGS approach (ie, BaP, ENU, PRC and TEM) were all investigated using the maximum tolerated dose for a 28-day exposure regimen. This was explained in the original manuscript (now lines 465-466 on page 21 of the revised manuscript). Thus, we could not test higher doses in an attempt to increase the mutagenic response of those chemicals that produced a weaker response with respect to that of BaP because they would have resulted in the death of the animals.

As requested, we are now reporting the standard deviation in Supplementary Table 2.

Minor issues

1. The order of the substitution pattern on the horizontal axis of the mutation spectrum graph should be rearranged to that of signature graph.

We have done as requested. The order of substitutions in Figure 2 and Supplementary Figure S2 is now the same as in the COSMIC signatures (eg, Figure 5).

2. I recommend that the similarity of signatures should be discussed based on cosine similarity, not Pearson's coefficient, in the manner similar to that employed in human cancer research.

We have done as requested. Using the cosine similarity values did not change the results that were obtained with the Pearson's coefficient.

3. In Discussion, the information on "background COSMIC signature" should be deleted. The concept of "background signature" in human cancer is scientifically inappropriate.

It was not our intention to suggest that the background signature is involved in carcinogenesis. Rather, as it was described in the original manuscript (now lines 203-206 on page 10 of revised manuscript), we contend that when mutations are analyzed at a short time after the end of the exposure, the contribution of spontaneous mutations to the mutation pattern of a mutagen cannot be ignored, especially for those mutagens that produce a weak response. Indeed, we show that when we include the background signature, the correlation between the mutation pattern of an agent and the reconstructed signature is improved and, furthermore, the use of the background signature removed spurious associations with irrelevant signatures (see lines 209-213 on page 10 of revised manuscript). This is line with the approach used by Kucab et al (2019) who also pointed out that "*Common culture reagents could generate mutational signatures and potentially confound interpretation of mutagen treatments*" and "*To ensure systematic, robust signature discovery, we defined the ubiquitous background signature based on control samples and ensured that the total number of mutations for any given treatment was significantly greater than the controls*". We have revised text to clarify the purpose and use of the background signature (see lines 268-271 on page 13 of the revised manuscript).

Reviewers' comments:

Reviewer #3 (Remarks to the Author):

Major issues

1. I do not agree with the arguments presented by the authors. Since the mechanisms of mutagenesis are not simple, the authors should demonstrate the rationale of their results in conjunction with the decomposition of mutational signatures. I think this manuscript fails to present evidence supporting the results depicted in Fig. 6. Therefore, as requested in the previous revision, the authors should explain the mechanisms with biological data-based evidence, allowing the interpretation of their decomposition results. For example, if the authors insist that the signature of UVB is composed of 33% SBS2 (APOBEC/AID) and 22% SBS7 (Sunlight), they should present the rationale and evidence that UVB truly damages DNA with these 2 separate mechanisms. This is also applicable to the results of other mutagens evaluated in this study. For example, for NDMA, the authors should demonstrate, based on experimental data, what parts of NDMA-adducts cause SBS11 or SBS30, respectively. Otherwise, it seems that the authors only performed simple additions of mutational signatures. In addition, I am afraid that there are several points where authors fail to rationally interpret their results. First, the authors have mentioned that "This may indicate that SBS2 is a signature of UVB, while SBS7 is a signature of other mutagenic components of sunlight" on line 216 of page 10 of the revised manuscript. However, according to the COSMIC DB, SBS2 is a signature attributed to the activity of AID/APOBEC family. I cannot understand why the authors concluded that SBS2 would be a signature of UVB. Furthermore, as the authors have written, if SBS2 was a signature of UVB and SBS7 was a signature of other components of sunlight, why would SBS7 still be included as part of the UVB signature in this study, as shown in Fig. 6? It seems logically inconsistent. Based on these examples, I think that decomposing the signatures of single chemicals with `deconstructSigs` based on data in human cancer could be a misleading way of understanding chemical mutagenesis. As mentioned in the previous review comments, the authors need to explain the mechanism and show biological experiment-based evidence.

As indicated by the authors, the results of the decomposition could change based on the repertoire of COSMIC signatures. Therefore, I do not agree with the adoption of this analysis unless the authors present biological experiments-based evidence to justify their results. Although the authors insist that their results were reliable because both the cosine similarity and the `deconstructSigs` approaches showed good concordance, I do not agree with this explanation. For example, if a signature that can explain 50% of a mutagen's signature existed, this signature should be accounted for by the `deconstructSigs` approach, but they do not necessarily show a high similarity based on the cosine similarity approach. I think it is very difficult to judge whether the current DB is sufficient for the decomposition of chemical mutagens signature. Therefore, as previously requested, the reliability of the decomposition results should be confirmed by biological experiments.

2. Irrespective of whether the signature pattern similar to mutagens' signatures can be identified from COSMIC DB, I worry about the reliability of the chemical-specific signatures of some mutagens in this study. As the authors have indicated in Fig. 6 and Supplementary Fig. 5, almost all mutagens showed substantial similarity to the pattern of the control signature. Since it is unlikely that all these mutagens share the same common mechanism, I am afraid that these data indicate that the number of mutations was insufficient to elucidate chemical-specific signatures of these mutagens. I recommend the authors to increase the number of detected mutations until this concern is resolved.

COMMSBIO-19-1762A

Beal et al – Sequencing chemically induced mutations in the MutaMouse lacZ reporter gene identifies human cancer mutational signatures.

Responses to reviewers

We appreciate the opportunity to revise the manuscript to address the additional comments provided by the reviewers. We believe that the new analyses that were requested strengthen our results, and thus, the revised manuscript is improved because of the input received. Below we provide our responses to the reviewers' comments. The line numbers in our responses refer to the line numbers in the version of the revised manuscript with track changes.

Associate Editor's comments,

We are interested in the possibility of publishing your study in Communications Biology, but would like to consider your response to these concerns in the form of a revised manuscript before we make a final decision on publication.

Specifically, R4 recommended to rephrase some statements and add some discussion points. R4 also suggested some basic computation ways to prove the robustness of the technique, which I think can be addressed.

As per communication with the Associate Editor, we have focused our responses and on revising the manuscript to address the comments provided by Reviewer 4. We agree that these were excellent suggestions to increase confidence in the data analysis and interpretation. Thus, we begin by answering the comments from this reviewer. However, we have evaluated the further comments received from Reviewer 3 and attempted to address these comments as much as we could.

Reviewers' comments:

Reviewer #4 (Remarks to the Author):

COMMSBIO-19-1762A

I was asked to comment independently on the rebuttal. I think the authors have done a solid effort in addressing several points raised. I have some recommendations.

We thank the reviewer for acknowledging that we had done a solid effort in addressing the comments that were received during the previous round of review. We also thank the reviewer for the recommendations on how to improve the manuscript. Below we provide

our responses to the comments. Note that we begin with addressing the last suggestion made by Reviewer 4 because it impacts the other suggestions as well.

4. The authors could use another mutation signature clustering algorithm besides deconstructSigs to convince the reader that their signals are robust and not artefacts of a particular approach. **As suggested by the reviewer, we have used a second mutation signature clustering algorithm to determine which COSMIC signatures may contribute to the mutation profile of each of the agents that we have investigated. Analyses conducted with MutationalPatterns, a recently developed algorithm (Blokzijl et al, 2018, Genome Medicine 10:13) with additional functionalities with respect to deconstructSigs, showed that the two algorithms produced remarkably similar results, both in terms of identified COSMIC signatures and % contribution (see Figure 6 and Supplementary Table S4). This suggests that the contribution of the selected COSMIC signatures to the mutation profile of an agent is not an artefact of a particular approach.**

Because of the inclusion of results from a second algorithm, we have modified our criteria for considering a COSMIC signature to be associated with the mutation profile of an agent by requiring that it must:

- 1) be detected by both deconstructSigs and MutationalPatterns;**
- 2) contribute at least 20% to the mutation profile;**
- 3) have a cosine value greater than 0.5 with the mutation profile of the agent.**

Note that to assure that we consider only those SBS signatures that have a strong association with the mutagenic mutation profiles, we are using a 20% contribution cut off. This is twice what is recommended by Blokzijl et al 2018. Furthermore, as suggested by the reviewer in a different comment, we present an analysis of how the number of 'detected' signatures decrease with increasing stringency parameters (eg, higher cosine value) to provide an idea of the robustness of the association of a given signature with a particular mutational profile. This is presented in lines 306-313 on page 12 of the revised manuscript and Supplementary Table S5.

Inclusion of the MutationalPatterns analysis required changing Figure 1, Figure 6, the addition of Supplementary Tables S4 and S5, Supplementary Figure S7 and many text changes that are too numerous to completely list here. However, we are providing a copy of the manuscript that shows all changes that were made in the text. Some of the changes are also identified in the responses below.

1. tone down the manuscript to address the overall concerns. Maybe basic things like "identify" vs "suggest" in the title and abstract would help. Also, make sure that the caveats list in the discussion section is prominent and covers the points raised.

As suggested, we have revised the title of the manuscript and toned down some of the statements abstract and throughout the manuscript. See for example:

Lines 31, 35 and 37 in the revised abstract.

Line 219, page 10; line 344, page 14; and line 516, page 20 of revised manuscript.

Also, there is an entire paragraph in the discussion (lines 525-548 on pages 20 and 21 of the revised manuscript) about the limitations of our study.

2. some of the points raised could be addressed with basic computation. For example, the authors could attempt to simulate (resample) mutation data and prove that their signal is robust.

This is an excellent idea and we have done what the reviewer suggested. First, we randomly downsampled the mutation data by 50% and demonstrated that there are minimal changes in the signatures identified (see Supplementary Figure S6). Secondly, we shuffled the mutation data randomly within trinucleotide mutation groups and show that reconstructions of the random permutations of our data do not lead to similar conclusions, nor do they lead to accurate reconstructions of the mutational profiles we analyzed (see Supplementary Figure S7). These additional tests demonstrate both the robustness of the signal compared to background noise, and the biological significance of the signature results as they relate to the mutagen being tested. These changes are reflected in lines 306-313 on page 12 of the revised manuscript.

3. On the same note, the authors could attempt to change parameters of the signature analysis method and derive fewer signatures. That would also demonstrate robustness of their technique. **As indicated in our response to comment 4 above, we have done what the reviewer has requested (see Supplementary Table S5). Changing these parameters did result in the elimination of three SBS signatures; however, there are still six SBS that are associated with the mutation profiles of some of the agents investigated here. Importantly, the known etiology of these SBS signatures are consistent with the known mutagenic mechanism of the agents they are associated with. Thus, we believe that these results are biologically relevant and not spurious methodological findings.**

Reviewer #3 (Remarks to the Author):

Major issues

1. I do not agree with the arguments presented by the authors. Since the mechanisms of mutagenesis are not simple, the authors should demonstrate the rationale of their results in conjunction with the decomposition of mutational signatures. I think this manuscript fails to present evidence supporting the results depicted in Fig. 6. Therefore, as requested in the previous revision, the authors should explain the mechanisms with biological data-based evidence, allowing the interpretation of their decomposition results.

It is clear that there remains a divergence of opinion with this reviewer. The major request from this reviewer is to conduct additional experiments to provide in his/her words “...biological experiment-based evidence”. We respectfully disagree with the reviewer that we have failed to “....demonstrate the rationale of results in conjunction with the decomposition of mutational signatures.”

We think that the evidence is provided by the fact that the molecular mechanisms that underly the COSMIC signatures that are associated with specific mutagens in our study are consistent with their known mutagenic mechanisms. For example, the mutational mechanisms associated with SBS2 and SBS7a are consistent with those expected to be operating after exposure to sunlight and UV radiations; the same is true for SBS 11 and SBS 30 and NDMA.

Other findings are also consistent with what other groups have found using different methodologies and approaches. For example, multiple experiments *in vitro* and *in vivo* have shown that the pattern of mutations induced by BaP resembles that of SBS 4. However, SBS 4 alone is not sufficient to explain 100% of the mutations induced by BaP, otherwise the cosine value would approach 1; in our case the cosine value between SBS 4 and the mutation profile of BaP is 0.81. The logical conclusion is that other mutagenic processes are involved and deconstructing the mutation profile into multiple SBS signatures is an attempt to identify these additional mutational mechanisms. Our results suggest that the mutagenic mechanism responsible for SBS 39 are contributing to the types of mutations induced by BaP. This would require experimental confirmation and is consistent with the stated purpose of our study:

“...argue that analysis of COSMIC signatures observed in exposed animals can be used to generate or test hypotheses of mutagenic mechanisms associated with human mutational signatures of unknown etiology.” (lines 121-123, pages 5-6 of revised manuscript).

“...our approach may be used to: (i) test if TGR mutation patterns support hypotheses that COSMIC signatures are attributed to particular mutagenic exposures, and (ii) generate hypotheses about the mutagenic mechanisms underlying human cancers through identifying enriched COSMIC signatures in TGR mutation patterns.” (lines 339-342, pages 14 of revised manuscript).

“Overall, these results demonstrate that lacZ transgene sequence data may be used, in conjunction with established mutation signatures derived from COSMIC cancer data sets, to test the hypothesis that a given class of mutagenic agents is linked with specific human cancers. Moreover, COSMIC signature mining based on TGR mutation data sets can be used to generate new hypotheses regarding the mutagenic mechanisms associated with human cancers.” (lines 516-518, page 20 of revised manuscript).

For example, if the authors insist that the signature of UVB is composed of 33% SBS2 (APOBEC/AID) and 22% SBS7 (Sunlight), they should present the rationale and evidence that UVB truly damages DNA with these 2 separate mechanisms. This is also applicable to the results of other mutagens evaluated in this study.

We have already pointed out in the manuscript (now line 404 on page 16 of the revised manuscript) that cytosine deamination is one of the major mutagenic mechanisms associated with SBS 2 and that it is induced by UV radiation: “Mechanistically, cytosine deamination is accelerated by UV exposure⁵⁵; thus, it is possible that we observed SBS 2 in this study because of UV-dependent cytosine deamination.” This is extensively published in the literature and additional experiments are thus not required. Regarding SBS7a, the COSMIC website states that “SBS7a may possibly be the consequence of just one of the two major known UV photoproducts, cyclobutane pyrimidine dimers or 6-4 photoproducts”. Thus, the presence of these two SBS signatures in the mutation profile of Sunlight and UV is consistent with the well-established and accepted mutagenic mechanisms of sunlight radiations. The latter provides an excellent hypothesis for future work outside the scope of this paper.

For example, for NDMA, the authors should demonstrate, based on experimental data, what parts of NDMA-adducts cause SBS11 or SBS30, respectively. Otherwise, it seems that the authors only performed simple additions of mutational signatures.

It is apparent from multiple rounds of commenting that this reviewer is highly critical of the use of deconstructSigs to identify COSMIC signatures that contributed to the mutation profile of the agents that were tested in our study. However, no concern was expressed on the analysis shown in Figure 4 where the mutation profile of each agent was compared against each individual SBS. This analysis shows that the mutation profile of NDMA has a cosine value of 0.83 and 0.85 with SBS 11 and SBS 30, respectively. A reasonable conclusion from this result is that both SBS 11 and SBS 30 contribute to the mutation profile of NDMA. In addition, DeconstructSigs, and now MutationalPatterns, both suggest that these two SBS signatures are involved, and that SBS 30 may be contributing more than SBS 11. We agree with the reviewer that it would be interesting to discriminate in follow-up studies which NDMA-adducts are associated with SBS11 and SBS30, but it is beyond the scope of this study (which is proof of concept and hypothesis generating).

In addition, I am afraid that there are several points where authors fail to rationally interpret their results. First, the authors have mentioned that “This may indicate that SBS2 is a signature of UVB, while SBS7 is a signature of other mutagenic components of sunlight” on line 216 of page 10 of the revised manuscript. However, according to the COSMIC DB, SBS2 is a signature attributed to the activity of AID/APOBEC family. I cannot understand why the authors concluded that SBS2 would be a signature of UVB. Furthermore, as the authors have written, if SBS2 was a signature of UVB and SBS7 was a signature of other components of sunlight, why would SBS7 still be included as part of the UVB signature in this study, as shown in Fig. 6? It seems logically inconsistent.

The reviewer misunderstood how to read Figure 6. In that Figure, we were reporting the contribution of all signatures that were identified by deconstructSigs. However, only the signatures that passed our selection criteria for association with the mutation profile of an agent were highlighted in dark red. For sunlight, both SBS2 and SBS7b were highlighted in dark red, while only SBS2 was highlighted in dark red for UV. We acknowledge that our writing was not clear regarding this point and in this revised version of the manuscript we have rewritten the text (see lines 235-264 on pages 10-11 of the revised manuscript) and modified Figure 6.

Based on these examples, I think that decomposing the signatures of single chemicals with deconstructSigs based on data in human cancer could be a misleading way of understanding chemical mutagenesis. As mentioned in the previous review comments, the authors need to explain the mechanism and show biological experiment-based evidence.

Please see our first response to this reviewer.

As indicated by the authors, the results of the decomposition could change based on the repertoire of COSMIC signatures. Therefore, I do not agree with the adoption of this analysis unless the authors present biological experiments-based evidence to justify their results. Although the authors insist that their results were reliable because both the cosine similarity and the deconstructSigs approaches showed good concordance, I do not agree with this explanation. For example, if a signature that can explain 50% of a mutagen’s signature existed, this signature

should be accounted for by the deconstructSigs approach, but they do not necessarily show a high similarity based on the cosine similarity approach. I think it is very difficult to judge whether the current DB is sufficient for the decomposition of chemical mutagens signature. Therefore, as previously requested, the reliability of the decomposition results should be confirmed by biological experiments.

As previously stated, now lines 540-548 on page 21 of the revised manuscript “Finally, we failed to identify COSMIC signatures contributing to the mutation profile of some of the agents tested.We suggest that: (1) COSMIC signatures do not yet capture all possible mutagenic mechanisms and are insufficient to appropriately decompose all mutagenic signatures; or (2) there is yet an insufficient number of cancers in the COSMIC database where these agents play a role in the carcinogenic process”, we acknowledged that the current database may not be sufficient for decomposition of all chemical signatures but this is not a fault of our study.

2. Irrespective of whether the signature pattern similar to mutagens' signatures can be identified from COSMIC DB, I worry about the reliability of the chemical-specific signatures of some mutagens in this study. As the authors have indicated in Fig. 6 and Supplementary Fig. 5, almost all mutagens showed substantial similarity to the pattern of the control signature. Since it is unlikely that all these mutagens share the same common mechanism, I am afraid that these data indicate that the number of mutations was insufficient to elucidate chemical-specific signatures of these mutagens. I recommend the authors to increase the number of detected mutations until this concern is resolved.

The fact that the control signature is associated with the mutation profile of several agents does not mean that they have the same common mechanism as the reviewer states. It means that the number of induced mutations is insufficient to ‘hide’ the contribution coming from spontaneous mutations. This is especially true when the mutagenic response of an agent is weak and increasing the number of sequenced mutations will not necessarily alleviate this. For example, as shown in Supplementary Table 2, the background mutant frequency in bone marrow is 5.71×10^{-5} , while the induced mutant frequency for TEM is 9.18×10^{-5} . This means that even if we were to sequence 1,000 additional mutations, about 50% of them would be spontaneous and not induced by TEM. Thus, increasing the sample size would not eliminate the association of the control signature with the mutation pattern of TEM. For these reasons, we think that the suggestion from the reviewer to increase the number of sequenced mutations is not biologically sound and would represent an unnecessary use of resources and, most importantly, animals.

We have added text in line 351-357 on page 14 of the revised manuscript to clarify that the presence of the control signature in the mutation profile of several agents does not mean that they operate through a common mechanism.

REVIEWERS' COMMENTS:

Reviewer #4 (Remarks to the Author):

The authors have addressed my comments reasonably well and I am happy to see that the additional computational approaches helped refine their signals.

There is one additional important caveat that the authors should emphasize in a few places in their manuscript. They currently investigate a fairly small number of mutations in the mutamouse model (a few thousand SNVs), while mutation signatures in human tumors are inferred from hundreds of thousands or millions of mutations. This is important to keep in mind while critically evaluating their findings.

COMMSBIO-19-1762B

Beal et al – Chemically induced mutations in a MutaMouse reporter gene informs mechanisms underlying human cancer mutational signatures.

Responses to Reviewer #4

The authors have addressed my comments reasonably well and I am happy to see that the additional computational approaches helped refine their signals.

We thank the reviewer for the positive comments and acknowledge that the additional work that this reviewer had requested during the previous round of review has undoubtedly made the manuscript better.

There is one additional important caveat that the authors should emphasize in a few places in their manuscript. They currently investigate a fairly small number of mutations in the mutamouse model (a few thousand SNVs), while mutation signatures in human tumors are inferred from hundreds of thousands or millions of mutations. This is important to keep in mind while critically evaluating their findings.

We agree with the reviewer. We had already pointed this out in Materials and Methods in the “Signature Analyses” section. Specifically, we write: *”To determine which of the COSMIC mutation signatures best explained the observed lacZ mutant pattern, the COSMIC mutation signature weights, which are derived from human mutation data, were first normalized to lacZ trinucleotide frequencies. This was done using the ratio of trinucleotide frequencies in lacZ to the trinucleotide frequencies in the human genome (Figure 3; the normalized signatures are shown in Supplementary Figure S3 and the raw numbers in Supplementary Material). Analysis was done this way (as opposed to converting lacZ mutation data themselves to human trinucleotide frequencies) because the COSMIC signatures are based on a much larger database, and therefore, represent a more robust signal with less variance.”* **We have now added a similar statement in the section on identifying COSMIC signatures in the Results. Specifically, we state: “Because the COSMIC database is based on a much larger dataset of mutations than the available lacZ mutations, we first divided each trinucleotide frequency in the lacZ transgene (Figure 3) by the respective human genome frequencies (hg38)...”** with the underline indicating the added text.